# PLUTO: A BENCHMARK FOR EVALUATING EFFICIENCY OF LLM-GENERATED HARDWARE CODE

## ABSTRACT

Large Language Models (LLMs) are increasingly used to automate hardware design tasks, including the generation of Verilog code. While early benchmarks focus primarily on functional correctness, efficient hardware design demands additional optimization for synthesis metrics such as area, delay, and power. Existing benchmarks fall short in evaluating these aspects comprehensively: they often lack optimized baselines or testbenches for verification. To address these gaps, we present Pluto, a benchmark and evaluation framework designed to assess the efficiency of LLM-generated Verilog designs. Pluto presents a comprehensive evaluation set of 114 problems with self-checking testbenches and multiple Pareto-optimal reference implementations. Experimental results show that state-of-the-art LLMs can achieve high functional correctness, reaching 78.3% at pass@1, but their synthesis efficiency still lags behind expert-crafted implementations, with area efficiency of 63.8%, delay efficiency of 65.9%, and power efficiency of 64.0% at eff@1. This highlights the need for efficiency-aware evaluation frameworks such as Pluto to drive progress in hardware-focused LLM research.

## 1 INTRODUCTION

Large Language Models (LLMs) are beginning to reshape hardware design by automating key steps in hardware design workflows, including Verilog code generation Thakur et al. (2023a;b); Liu et al. (2023a), optimization Yao et al. (2024); Guo & Zhao (2025), verification Qiu et al. (2024a), debugging Tsai et al. (2024), high-level synthesis Xiong et al. (2024), and post-synthesis metric estimation Abdelatty et al. (2025). While these advances highlight the potential of LLMs in hardware design, most research has focused on functional correctness of generated designs, with little attention to design quality metrics such as area, delay, and power.

In hardware design, the quality of Verilog code is not determined solely by functional correctness. Designs typically undergo logic synthesis, where Verilog code is mapped to gate-level implementations in a target technology. This process exposes critical efficiency metrics, such as silicon area, timing delay, and power consumption, that directly impact manufacturability and performance. Unlike software code, where correctness and execution speed often suffice, hardware code quality is inherently tied to these post-synthesis metrics.

In order to evaluate the functional correctness of LLM-generated Verilog code, several benchmarks have been proposed including VerilogEval Liu et al. (2023a) and RTLLM Lu et al. (2024). Recent efforts, including RTLRewriter Yao et al. (2024), ResBench Guo & Zhao (2025), GenBen Wan et al. (2025), and TuRTLe Garcia-Gasulla et al. (2025), have begun to evaluate quality of LLM-generated hardware code in terms of post-synthesis metrics. However, these benchmarks face key limitations:

- **Absence of Optimal Ground Truth Solutions** True efficiency should be measured against implementations that are explicitly optimized for specific objectives such as silicon area, delay, or power consumption. Prior studies rely on canonical solutions from VerilogEval and RTLLM as reference solutions. Our analysis shows that these solutions are not the most optimal in terms of post-synthesis metrics.

- **Lack of Clock Latency Agnostic Testbenches** Many common optimization patterns, such as register pipelining, resource sharing, or FSM restructuring, introduce variations in clock-cycle latency between the optimized and unoptimized designs. To support fair evaluation,

testbenches must be self-checking and tolerant of different latency requirements. Existing benchmarks, however, assume identical latency between the reference model and the design under test, making them unsuitable for efficiency benchmarking.

In order to address these limitations, we introduce *Pluto*, the first benchmark designed to evaluate both *functional correctness* and *synthesis efficiency* of LLM-generated Verilog code. Our contributions are as follows:

- **Per-Metric Ground Truth Optimal Solutions.** We provide a suite of 114 problems where each is optimized for area, delay, and power separately, yielding Pareto-front optimal solutions. Our ground truth solutions are significantly more efficient than canonical solutions in RTLLM and VerilogEval.

- **Optimization-Aware Testbenches.** Each problem is accompanied by clock-cycle agnostic testbenches that accommodate varying latency requirements, ensuring robust evaluation of different optimization patterns.

- **Comprehensive Evaluation.** We adapt the *eff@k* metric introduced in Qiu et al. (2024b) to measure the efficiency of hardware designs. Our extended metric is a three-dimensional vector that evaluates LLM-generated code across multiple objectives: area, delay and power.

## 2 RELATED WORK

**Software Code Benchmarks** Large Language Models (LLMs) have been extensively studied for code generation across both software and hardware domains, with most early benchmarks focusing primarily on functional correctness rather than efficiency. In software, works such as Mercury Du et al. (2024) and ENAMEL Qiu et al. (2024b) move beyond correctness to explicitly evaluate runtime efficiency of LLM-generated programs. The Mercury Du et al. (2024) benchmark contains LeetCode style problems. Each problem is accompanied by an expert-written solution that represents the most optimal implementation in terms of run-time efficiency. ENAMEL Qiu et al. (2024b) also introduces a Python benchmark to evaluate the run-time efficiency of LLM-generated code.

**Hardware Code Benchmarks** In hardware design, early work on LLM-generated Verilog emphasized functional correctness. VerilogEval Liu et al. (2023a) only evaluates whether the LLM generated code passes the testbench check, while RTLLM Lu et al. (2024) additionally checks if the generated code is synthesizable. More recent efforts have shifted toward assessing and improving the efficiency of LLM-generated designs, which can be categorized into two main categories: *Specifications-to-Efficient-Verilog* where the LLM is tasked with translating natural language instruction to optimized Verilog code directly, and *Unoptimized-Verilog-to-Optimized-Verilog*, where the LLM is tasked with rewriting an unoptimized Verilog code to optimized Verilog code.

In the *Specifications-to-Efficient-Verilog* formulation, the LLM is prompted with a natural language problem description and directly generates optimized Verilog. Benchmarks, such as GenBen Wan et al. (2025), TuRTLe Garcia-Gasulla et al. (2025), evaluate these generations in functional correctness, synthesizability, and post-synthesis metrics such as area, delay, and power. However, it relies on VerilogEval problems as ground truth. These reference designs are not necessarily optimized for power, performance, or area, and thus do not represent true Pareto-optimal solutions. ResBench Guo & Zhao (2025) also does not define any gold-standard or reference-optimal implementations, which makes it difficult to quantitatively assess how close the generated solutions are to ideal results.

The *Unoptimized-Verilog-to-Optimized-Verilog* setting provides the LLM with a functionally correct but unoptimized Verilog implementation and asks it to produce a more efficient version. RTL-Rewriter Yao et al. (2024) enhances this with retrieval-augmented generation and feedback through the synthesis loop. However, RTLRewriter lacks associated testbenches, making it unsuitable for assessing the functional correctness of the generated code.

As summarized in Table 1, *Pluto* is the first benchmark to offer per-metric optimization, providing separate expert-optimized reference designs for area, delay, and power. This enables targeted, metric-specific evaluation of LLMs, an aspect missing from prior benchmarks.

Table 1: Comparison of prior software and hardware code generation benchmarks. *Pluto* addresses key limitations by enabling metric-specific optimization with three reference implementations per problem, each optimized for area, delay, or power.

| Benchmark | Language | Functionality | Synthesizability | Efficiency | Per-Metric Optimisation | Tasks |
|---|---|---|---|---|---|---|
| HumanEval | Python | ✓ | − | ✗ | ✗ | 164 |
| Mercury | Python | ✓ | − | ✓ | ✗ | 256 |
| ENAMEL | Python | ✓ | − | ✓ | ✗ | 142 |
| VerilogEval | Verilog | ✓ | ✗ | ✗ | ✗ | 156 |
| RTLLM | Verilog | ✓ | ✓ | ✗ | ✗ | 30 |
| RTLRewriter | Verilog | ✗ | ✓ | ✓ | ✗ | 95 |
| ResBench | Verilog | ✓ | ✓ | ✗ | ✗ | 56 |
| GenBen | Verilog | ✓ | ✓ | ✗ | ✗ | 300 |
| TuRTLe | Verilog | ✓ | ✓ | ✗ | ✗ | 223 |
| CVDP | Verilog | ✓ | ✓ | ✗ | ✗ | 783 |
| **Pluto (Ours)** | Verilog | ✓ | ✓ | ✓ | ✓ | **114** |

# 3 PLUTO BENCHMARK

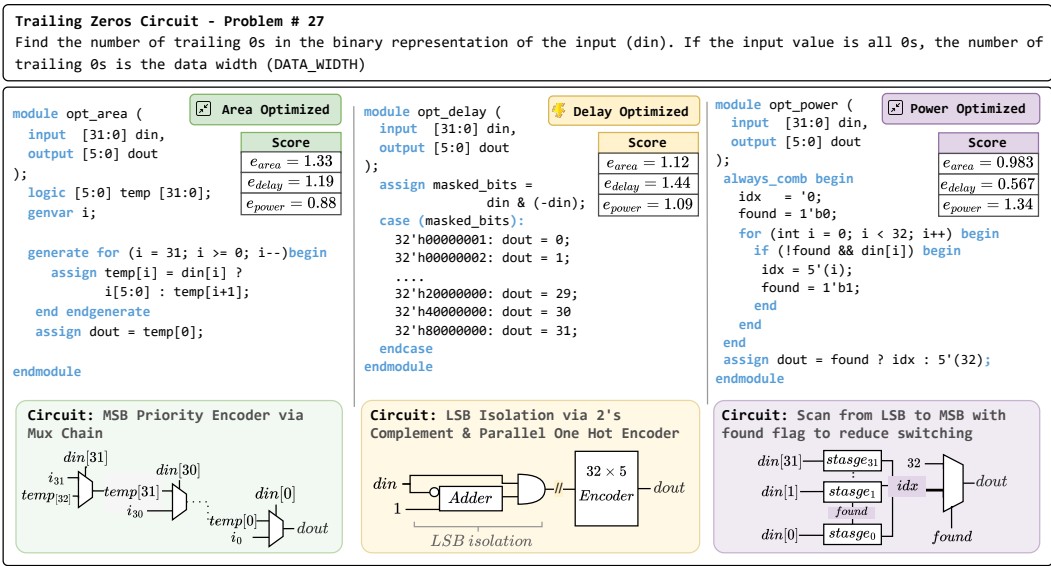

Figure 1: Overview of the *Pluto* benchmark on the trailing zeros detection task. We show three reference implementations optimized for different synthesis metrics compared to the unoptimized baseline: (left) area, using a mux-based priority encoder, reducing area by 33%; (center) delay, using an LSB isolation circuit with a parallel one-hot encoder, reducing delay by 44%; and (right) power, using an LSB-to-MSB scanning method with early termination, reducing total power by 34%. See Appendix. A.1 for unoptimized baseline and self-checking testbench.

## 3.1 DATA CONSTRUCTION

To enable a comprehensive evaluation of synthesis efficiency for LLM-generated hardware code, we construct the *Pluto* evaluation set, which contains a diverse collection of high-quality digital design problems spanning a broad range of difficulties. Specifically, we curated 114 problems from various publicly available sources, including open-source hardware projects, educational platforms such as ChipDev ChipDev (2025), a LeetCode-inspired platform for practicing Verilog coding, and prior benchmark suites such as RTLRewriter Yao et al. (2024), RTLLM Lu et al. (2024), and Ver-

ilogEval Liu et al. (2023b). Each problem is specified by a high-level description outlining the functional requirements, together with a baseline unoptimized Verilog implementation.

The problem set covers a wide spectrum of tasks in digital logic design, ranging from arithmetic units and control circuits to sequential state machines. To systematically capture variation in design complexity, we adopt ChipDev's difficulty annotations and classify problems into three levels: easy, medium, and hard. These labels reflect the intrinsic challenge of translating the textual description into a correct Verilog implementation, thereby providing a principled way to distinguish between problems of different complexity.

Importantly, the resulting collection balances accessibility with challenge: many problems that appear straightforward can nonetheless expose substantial differences in synthesis efficiency depending on the optimization strategies applied. The diverse composition of easy, medium, and hard tasks therefore enables a nuanced assessment of an LLM's ability to generate synthesis-efficient Verilog under varying constraints. In total, the 114 selected problems provide a representative and scalable testbed for benchmarking LLM-based Verilog efficiency.

Each problem instance in the *Pluto* benchmark includes the following components:

- **Prompt**: A natural language description of the hardware design task intended to guide the LLM-generation.
- **Module Header**: A fixed interface shared across all versions of the Verilog module to ensure consistency and comparability.
- **Unoptimized Verilog Code**: A baseline implementation used as the reference for testing.
- **Optimized Verilog Code**: Three distinct implementations with tradeoffs, each optimized by hand using design experts for a single metric: area, delay, or power.
- **Testbench**: A manually crafted, fully self-checking testbench that verifies functional equivalence between the unoptimized and any optimized design. These testbenches ensure full input space coverage and flag any mismatches during simulation. For sequential circuits, testbenches are clock-cycle agnostic, supporting latency differences introduced by optimizations such as pipelining or resource sharing.

All components in the evaluation set are manually developed. This ensures high quality and guarantees that the LLM under evaluation has not previously encountered any part of the dataset during training. In particular, the testbenches and optimized code serve as held-out ground truth references, providing an unbiased benchmark for assessing the efficiency and correctness of LLM-generated Verilog designs.

To illustrate the structure of problems in the *Pluto* evaluation set, Figure 1 presents the example of a trailing zeros detection circuit, categorized as an easy problem, along with its three metric-specific optimizations. As shown, each optimization achieves peak efficiency in its targeted metric, while performance in the remaining two metrics declines. This behavior emphasizes the inherent trade-offs across design objectives in hardware design and highlights the necessity of metric-specific optimization strategies.

## 3.2 Optimization Workflow

Each unoptimized design in the *Pluto* set is further refined through manual optimization by expert engineers to generate three distinct versions optimized separately for area, delay, and power. This workflow follows a systematic process that ensures both the correctness and the efficiency of the resulting designs. After applying metric-specific transformations, each optimized circuit is rigorously verified for functional correctness using Icarus Verilog Williams et al. (2002), supported by robust self-checking testbenches that guarantee equivalence with the unoptimized baseline. The optimized versions are then synthesized to confirm that improvements translate into measurable gains in area, timing, or power, thereby providing reliable performance baselines against which LLM-generated designs can be evaluated.

To understand how these efficiency gains are achieved, we visualize the optimization strategies applied across the dataset in appendix A.2. The strategies vary significantly depending on the target metric. For area, arithmetic optimizations and logic simplification are most commonly employed,

and FSM restructuring plays an important role in reducing redundant states and transitions. Delay improvements rely heavily on exploiting parallelism and restructuring control logic, often complemented by logic simplification and pipelining techniques that shorten the critical path. For power, it's reducing switching activity through register and logic optimizations, supported by techniques such as operand isolation, and clock gating to further suppress unnecessary toggling.

The distribution of strategies reveals that no single optimization technique dominates across all objectives. Instead, engineers select strategies tailored to the specific metric, reflecting the trade-offs inherent in digital design. As shown in Figure 2, this process results in consistent improvements across all *114* designs, with average reductions of 19.19% (SD=18.99%) in area, 21.96% (SD=20.99%) in delay, and 22.55% (SD=21.65%) in power. This highlights the importance of metric-specific approaches and provide a robust baseline for evaluating LLM-generated hardware code efficiency.

To further illustrate the impact of expert-driven optimization, Table 2 presents representative examples drawn from both RTLLM and VerilogEval. These case studies highlight how different strategies, such as arithmetic unit sharing, FSM encoding choices, and counter-based control logic, translate into concrete improvements across area, delay, and power. As shown, across both VerilogEval and RTLLM, expert-optimized designs consistently outperform baseline implementations. In particular, for RTLLM problems our expert-written solutions achieve average improvements of 18.75% (SD=14.55%) in area, 22.75% (SD=20.99%) in delay, and 20.43% (SD=21.65%) in power compared to their canonical solutions. For VerilogEval problems, the improvements average 10.46% (SD=14.40%) in area, 10.33% (SD=15.10%) in delay, and 13.61% (SD=18.86%) in power.

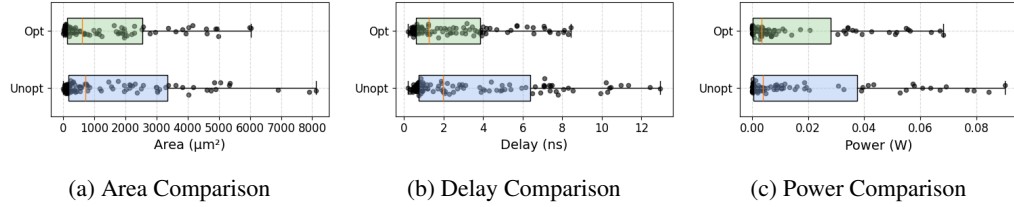

(a) Area Comparison    (b) Delay Comparison    (c) Power Comparison

Figure 2: Distribution of area, delay, and power across all 114 tasks in *Pluto* before and after manual metric-specific optimizations.

Table 2: A sample of benchmark problems from *Pluto* dataset. Our expert-optimized solutions (area, delay, power) are significantly more efficient than the baseline benchmark implementations. See Appendix A.3 for full problem implementation.

| ID | Source | Problem Description | Benchmark Solution | Expert Solution (Ours) |
|---|---|---|---|---|
| #60 | RTLLM | ALU for a 32-bit MIPS-ISA CPU with operations like {ADD, SUB, AND, OR, XOR, SLT, shifts, LUI}. | ALU implementation with case statement and parameterized opcodes. | **Area ↓ 26%:** Shared adder for arithmetic, simplified flag logic, and operand reuse
**Delay ↓ 26%:** Parallel datapaths with one-hot muxing
**Power ↓ 4%:** Operand gating and early zeroing for large shifts to cut switching activity |
| #68 | RTLLM | FSM detecting the sequence 10011 on a serial input stream with support for continuous and overlapping detection | States binary encoded, sequential next-state logic, and registered Mealy output | **Area ↓ 32%:** Casez-based transitions and direct output
**Delay ↓ 17%:** One-hot state encoding with pre-decoded inputs and Moore-style output
**Power ↓ 23%:** Compact binary encoding and casez-based transitions to cut toggling |
| #87 | VerilogEval | Module controls a shift register enable signal, $shift\_ena$ asserted for 4 clock cycles on reset, then remain 0 until the next reset | Uses explicit states with next-state logic to drive shift_ena | **Area ↓ 47%:** Minimized register width (2-bit counter) and compact comparator logic
**Delay ↓ 37%:** Wider counter (3 bits) to simplify comparison and reduce logic depth on the critical path
**Power ↓ 46%:** Small counter reused |
| #104 | VerilogEval | Conway's Game of Life with a $16 \times 16$ toroidal grid: each cell updates based on neighbor counts $n$ (live if $n = 3$ or $n = 2$ & alive) | Straightforward RTL with per-cell neighbor recomputation and sequential summing | **Area ↓ 19%:** Shared neighbor computations across rows, bitwise rotations for wraparound, less duplicate summations
**Delay ↓ 37%:** Parallel neighbor summation with carry-save adder tree and direct decode for 2 and 3
**Power ↓ 36%:** Reduced toggling via computation reuse |

## 3.3 EFFICIENCY METRICS

We use the *pass@k* Liu et al. (2023a) for measuring the functional correctness of LLM-generated Verilog code. The *pass@k* metric, defined in appendix. A.4, measures the percentage of problems for which at least one of the top-$k$ generated samples passes the self-checking testbench.

To evaluate the synthesis efficiency of functionally correct samples, we adapt the *eff@k* introduced in Qiu et al. (2024b) to Verilog code. First, we introduce the efficiency score $e_{i,j}$, defined in Eq. 1, which quantifies how close an LLM-generated design is to optimal ground truth implementation. In this equation, $\hat{R}_{i,j}$ denotes the reported synthesis metric (e.g., area, delay, or power) for the $j$-th sample of problem $i$, $T_{i,j}$ denotes an upper bound beyond which the design is considered inefficient, and $R_{i,j}$ denotes the optimal (lowest) known reference value for that metric. A score of 1 indicates that the sample exactly matches the optimal reference, while a score of 0 indicates that it exceeds the acceptable threshold or is functionally incorrect.

$$e_{i,j} = \begin{cases} \dfrac{\max(0,\, T_{i,j} - \hat{R}_{i,j})}{T_{i,j} - R_{i,j}}, & \text{if } n_{i,j} \text{ is correct} \\ 0, & \text{otherwise.} \end{cases} \quad (1)$$

$$\begin{aligned} \text{eff@}k &= \frac{1}{N} \sum_{i=1}^{N} \mathbb{E}_{J \subseteq \{1,\dots,n\},\, |J|=k} \left[ \max_{j \in J} e_{i,j} \right] \\ &= \frac{1}{N} \sum_{i=1}^{N} \sum_{r=k}^{n} \frac{\binom{r-1}{k-1}}{\binom{n}{k}}\, e_{i,(r)}. \end{aligned} \quad (2)$$

We then use the efficiency score $e_{i,j}$ for computing the *eff@k*, defined in Eq. 2 as the average of the best (i.e., highest) efficiency scores among the top-$k$ functionally correct samples for each problem. We use the unbiased estimator introduced in Qiu et al. (2024b) for computing *eff@k* which computes the expectation value over a random subset $J$ of code samples with size $K$.

## 4 EVALUATION RESULTS

We evaluate 18 large language models (LLMs) using our *Pluto* benchmark, which includes proprietary LLMs, general-purpose foundation models, code-specialized models, and Verilog-tuned models. To comprehensively assess efficiency-aware generation, we consider the two problem formulations in *Pluto*: translating unoptimized Verilog code into optimized implementations, and generating optimized code directly from natural-language specifications. For the first problem formulation, only instruction-tuned models are evaluated, as code completion models generally reproduce the unoptimized code without meaningful improvements.

### 4.1 MAIN RESULTS

Table 3 (a) reports the *pass@k* and *eff@k* metrics for the first problem formulation, where the task is to re-write unoptimized Verilog into more efficient implementations. Several trends emerge. First, in terms of functional correctness (*pass@k*), domain-tuned models such as VeriThoughts-Inst-7B and RTLCoder-DeepSeek-V1 achieve performance comparable to much larger foundational models like DeepSeek-Chat, demonstrating the benefit of Verilog-specific training. However, in terms of synthesis efficiency (*eff@k*), all models exhibit a noticeable drop relative to their *pass@k* scores. This gap underscores a common limitation: while LLMs can generate functionally correct Verilog, they struggle to match the Pareto-efficient expert baselines across area, delay, and power.

Table 3 (b) reports the *pass@k* and *eff@k* metrics for the second problem formulation, where models are tasked with translating natural language specifications into optimized Verilog implementations. This task is more challenging, and as a result, both *pass@k* and *eff@k* scores are consistently lower across all models. Similar to the first formulation, all models also exhibit lower *eff@k* values compared to their corresponding *pass@k* scores, underscoring the persistent difficulty of generating designs that are not only functionally correct but also synthesis-efficient. However, the relative gap between *pass@k* and *eff@k* is smaller in this setting compared to the first formulation. This is because specification-to-RTL translation is substantially harder: models often struggle to produce functionally correct code in the first place, which suppresses both correctness and efficiency scores.

Table 3: Evaluation results using *Pluto* for two problem formulations: **P1: Unoptimized-Verilog-to-Optimized-Verilog** and **P2: Specifications-to-Optimized-Verilog**. *pass@k* measures functional correctness, while *eff@k* measures efficiency across area, delay, and power.

(a) **P1: Unoptimized-Verilog-to-Optimized-Verilog**

| Model | pass@1 | pass@5 | pass@10 | eff@1 | | | eff@5 | | | eff@10 | | |
|---|---|---|---|---|---|---|---|---|---|---|---|---|
| | | | | Area | Delay | Power | Area | Delay | Power | Area | Delay | Power |
| GPT-3.5 | 0.325 | 0.517 | 0.594 | 0.271 | 0.296 | 0.282 | 0.462 | 0.491 | 0.450 | 0.540 | 0.568 | 0.520 |
| GPT-4o-mini | 0.506 | 0.705 | 0.751 | 0.469 | 0.476 | 0.467 | 0.662 | 0.677 | 0.639 | 0.714 | 0.744 | 0.687 |
| DeepSeek-Chat | 0.612 | 0.802 | 0.860 | 0.586 | 0.599 | 0.601 | 0.776 | 0.795 | 0.794 | 0.839 | 0.862 | 0.846 |
| Llama-3.3-70B-Instruct | 0.473 | 0.701 | 0.757 | 0.446 | 0.462 | 0.429 | 0.662 | 0.696 | 0.662 | 0.707 | 0.760 | 0.735 |
| Llama-3.1-8B-Instruct | 0.160 | 0.432 | 0.567 | 0.127 | 0.156 | 0.145 | 0.358 | 0.437 | 0.384 | 0.494 | 0.584 | 0.505 |
| Mistral-7B-Instruct-v0.2 | 0.106 | 0.318 | 0.453 | 0.078 | 0.094 | 0.100 | 0.244 | 0.296 | 0.301 | 0.358 | 0.446 | 0.427 |
| Mixtral-8x7B-v0.1 | 0.255 | 0.520 | 0.652 | 0.217 | 0.231 | 0.210 | 0.462 | 0.487 | 0.447 | 0.593 | 0.630 | 0.561 |
| starcoder2-15b-instruct-v0.1 | 0.659 | 0.960 | 0.988 | 0.611 | 0.633 | 0.591 | 0.879 | 0.924 | 0.871 | 0.913 | 0.952 | 0.904 |
| CodeLlama-70b-Instruct-hf | 0.576 | 0.905 | 0.956 | 0.522 | 0.541 | 0.523 | 0.842 | 0.876 | 0.824 | 0.903 | 0.925 | 0.878 |
| DeepSeek-Coder-33B | 0.783 | 0.963 | 0.997 | 0.638 | 0.659 | 0.640 | 0.902 | 0.927 | 0.883 | 0.942 | 0.960 | 0.927 |
| Qwen2.5-Coder-7B-Inst | 0.479 | 0.785 | 0.866 | 0.438 | 0.452 | 0.419 | 0.710 | 0.759 | 0.741 | 0.785 | 0.848 | 0.833 |
| yang-z/CodeV-QC-7B | 0.231 | 0.416 | 0.506 | 0.211 | 0.208 | 0.187 | 0.381 | 0.390 | 0.361 | 0.455 | 0.491 | 0.442 |
| RTLCoder-DeepSeek-V1 | 0.532 | 0.854 | 0.915 | 0.471 | 0.495 | 0.468 | 0.774 | 0.789 | 0.757 | 0.843 | 0.850 | 0.809 |
| VeriThoughts-Inst.-7B | 0.611 | 0.797 | 0.854 | 0.540 | 0.560 | 0.524 | 0.708 | 0.740 | 0.702 | 0.763 | 0.785 | 0.765 |

(b) **P2: Specifications-to-Optimized-Verilog**

| Model | pass@1 | pass@5 | pass@10 | eff@1 | | | eff@5 | | | eff@10 | | |
|---|---|---|---|---|---|---|---|---|---|---|---|---|
| | | | | Area | Delay | Power | Area | Delay | Power | Area | Delay | Power |
| GPT-3.5 | 0.239 | 0.395 | 0.471 | 0.225 | 0.235 | 0.225 | 0.373 | 0.390 | 0.381 | 0.439 | 0.469 | 0.468 |
| GPT-4o-mini | 0.391 | 0.533 | 0.591 | 0.360 | 0.377 | 0.363 | 0.475 | 0.520 | 0.495 | 0.532 | 0.572 | 0.551 |
| DeepSeek-Chat | 0.557 | 0.688 | 0.719 | 0.545 | 0.552 | 0.528 | 0.689 | 0.680 | 0.651 | 0.726 | 0.710 | 0.684 |
| Llama-3.3-70B-Instruct | 0.363 | 0.541 | 0.594 | 0.348 | 0.345 | 0.342 | 0.515 | 0.533 | 0.510 | 0.564 | 0.602 | 0.557 |
| Llama-3.1-8B-Instruct | 0.087 | 0.224 | 0.301 | 0.075 | 0.073 | 0.081 | 0.189 | 0.184 | 0.204 | 0.270 | 0.251 | 0.279 |
| Mistral-7B-Instruct-v0.2 | 0.030 | 0.108 | 0.164 | 0.024 | 0.015 | 0.024 | 0.078 | 0.067 | 0.088 | 0.112 | 0.117 | 0.134 |
| Mixtral-8x7B-v0.1 | 0.082 | 0.176 | 0.222 | 0.082 | 0.068 | 0.079 | 0.172 | 0.131 | 0.163 | 0.202 | 0.166 | 0.211 |
| starcoder2-15b-instruct-v0.1 | 0.243 | 0.454 | 0.512 | 0.226 | 0.249 | 0.220 | 0.429 | 0.466 | 0.409 | 0.489 | 0.525 | 0.458 |
| CodeLlama-70b-Instruct-hf | 0.212 | 0.446 | 0.532 | 0.202 | 0.207 | 0.194 | 0.418 | 0.440 | 0.437 | 0.498 | 0.535 | 0.524 |
| DeepSeek-Coder-33B | 0.257 | 0.429 | 0.482 | 0.231 | 0.254 | 0.246 | 0.387 | 0.424 | 0.421 | 0.446 | 0.490 | 0.468 |
| Qwen2.5-Coder-7B-Inst | 0.164 | 0.324 | 0.389 | 0.158 | 0.162 | 0.148 | 0.307 | 0.319 | 0.295 | 0.366 | 0.375 | 0.357 |
| code-gen-verilog-16b | 0.068 | 0.200 | 0.289 | 0.069 | 0.064 | 0.058 | 0.188 | 0.175 | 0.188 | 0.268 | 0.253 | 0.272 |
| yang-z/CodeV-CL-7B | 0.265 | 0.485 | 0.553 | 0.233 | 0.243 | 0.237 | 0.432 | 0.455 | 0.436 | 0.493 | 0.536 | 0.511 |
| yang-z/CodeV-QC-7B | 0.260 | 0.458 | 0.529 | 0.223 | 0.229 | 0.222 | 0.420 | 0.415 | 0.410 | 0.497 | 0.480 | 0.478 |
| yang-z/CodeV-All-QC | 0.175 | 0.317 | 0.374 | 0.150 | 0.162 | 0.144 | 0.297 | 0.304 | 0.256 | 0.368 | 0.379 | 0.284 |
| RTLCoder-DeepSeek-V1 | 0.203 | 0.400 | 0.480 | 0.177 | 0.199 | 0.184 | 0.345 | 0.404 | 0.361 | 0.417 | 0.499 | 0.430 |
| RTLCoder-Mistral | 0.199 | 0.347 | 0.418 | 0.185 | 0.188 | 0.188 | 0.316 | 0.332 | 0.329 | 0.381 | 0.411 | 0.395 |
| VeriThoughts-Inst.-7B | 0.216 | 0.336 | 0.398 | 0.211 | 0.206 | 0.207 | 0.316 | 0.330 | 0.329 | 0.367 | 0.394 | 0.393 |

## 4.2 ABLATION STUDIES

In addition to the Verilog code writing style, post-synthesis metrics are also influenced by external factors such as the synthesis tool employed, the target technology library, and the optimization sequence executed by the tool. To understand the robustness of our proposed benchmark and isolate the impact of these factors, we present two ablation studies that evaluate efficiency trends in the *Pluto* benchmark across different synthesis tools, optimization strategies, and technology libraries.

### 4.2.1 SYNTHESIS TOOL AND TECHNOLOGY AGNOSTICISM

In this experiment, we repeated synthesis runs for the three optimized reference implementations in *Pluto*, as well as the unoptimized baseline, using two distinct synthesis tools: Yosys Wolf et al. (2013), an open-source framework, and Cadence Genus cad, a commercial synthesis tool. To further evaluate generalizability, we also targeted two technology libraries representing different fabrication nodes: the SkyWater 130nm library Google and a 65nm TSMC library tsm. We then computed the efficiency score for each tool and library configuration by comparing each optimized implementation against the corresponding unoptimized baseline across area, delay, and power metrics. As shown in Figure 3, efficiency scores remain consistent across all synthesis tool and technology combinations. This demonstrates that *Pluto*'s optimization patterns deliver consistent tradeoffs across different synthesis tools and technology libraries.

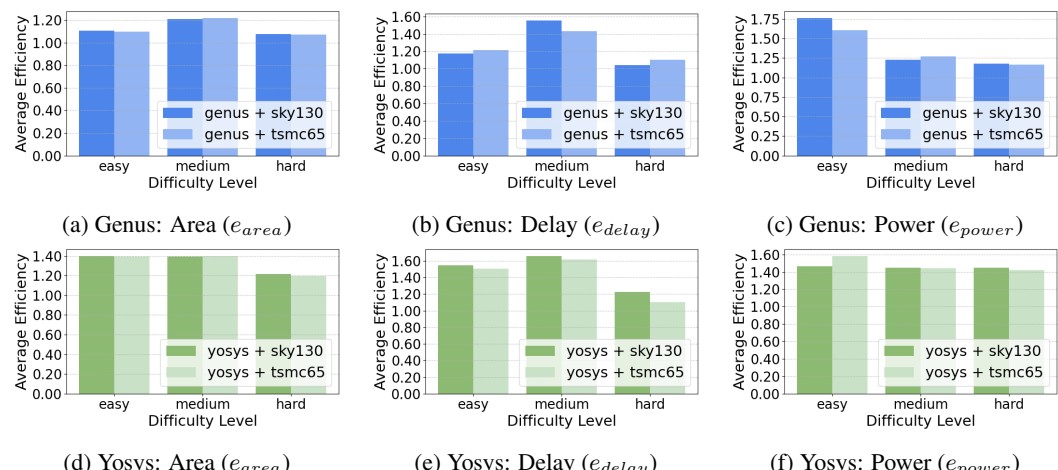

(a) Genus: Area ($e_{area}$)     (b) Genus: Delay ($e_{delay}$)     (c) Genus: Power ($e_{power}$)

(d) Yosys: Area ($e_{area}$)     (e) Yosys: Delay ($e_{delay}$)     (f) Yosys: Power ($e_{power}$)

Figure 3: Efficiency scores for area, delay, and power across all benchmark problems, using both Cadence Genus and Yosys with different technology libraries. Results show consistent efficiency trends across synthesis tools and technologies.

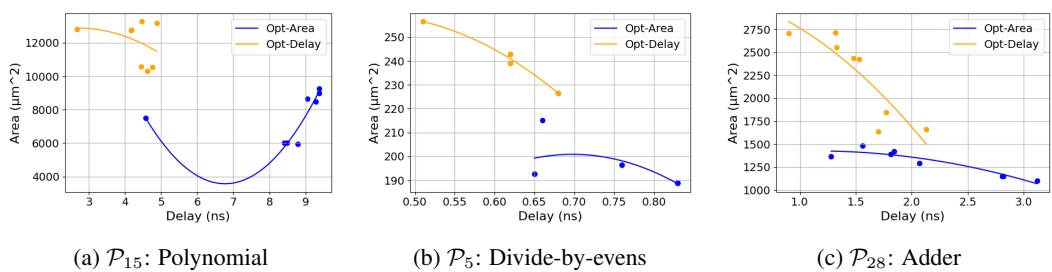

(a) $\mathcal{P}_{15}$: Polynomial     (b) $\mathcal{P}_5$: Divide-by-evens     (c) $\mathcal{P}_{28}$: Adder

Figure 4: Area–delay tradeoffs of three problems in the *Pluto* benchmark under different synthesis strategies. Each strategy corresponds to a distinct sequence of ABC logic synthesis commands.

### 4.2.2 SYNTHESIS OPTIMIZATION STRATEGIES

We also examine how different synthesis optimization strategies influence the post-synthesis metrics of *Pluto*'s optimized implementations. Synthesis tools allow designers to specify optimization directives that steer the tool's internal heuristics toward minimizing a particular metric while potentially sacrificing others. To study this effect, we synthesized selected problems from the *Pluto* benchmark under both area-optimized and delay-optimized optimization strategies. We used Yosys as our synthesis tool and targeted the SkyWater 130nm library. Within Yosys, logic optimization is carried out using the *ABC* framework Synthesis & Group (2024), which provides a collection of optimization heuristics that can be configured to emphasize different objectives such as area or delay minimization. Figure 4 illustrates the resulting Pareto fronts of area–delay trade-offs across three representative problems. As expected, delay-optimized code consistently achieves superior timing performance at the expense of larger area, whereas area-optimized code achieves lower area but incurs higher delays. These results confirm that synthesis settings primarily shift designs along the area–delay curve, while coding style remains the dominant factor, validating *Pluto*'s ability to capture design efficiency independent of synthesis optimization settings.

## 5 MULTI-OBJECTIVE OPTIMIZATION

*Pluto* also supports benchmarking LLMs for multi-objective optimization across area, delay, and power. For each problem, the individually optimized design variants represent distinct Pareto-optimal points. Together, they form a Pareto front that captures the trade-offs between these de-

Table 4: Multi-objective performance comparison across metric combinations using the P2 problem formulation (Specifications-to-Optimized-Verilog). Best metric set per model is bolded.

| Model | Area–Delay | | Area–Power | | Delay–Power | | Area–Delay–Power | |
|---|---|---|---|---|---|---|---|---|
| | pass@10 | eff@10 | pass@10 | eff@10 | pass@10 | eff@10 | pass@10 | eff@10 |
| GPT-4o-mini | 0.570 | 0.501 | **0.597** | **0.548** | 0.570 | 0.515 | 0.579 | 0.506 |
| Llama3.3-70B-Instruct | **0.649** | **0.615** | 0.605 | 0.577 | 0.640 | 0.600 | 0.605 | 0.567 |
| Qwen2.5-Coder-7B-Instruct | 0.395 | 0.358 | 0.395 | 0.364 | 0.421 | 0.393 | **0.430** | **0.402** |
| Mixtral-8x7B | 0.219 | 0.193 | 0.219 | 0.204 | **0.237** | **0.207** | 0.228 | 0.207 |
| StarCoder2 | 0.500 | 0.465 | 0.526 | 0.499 | **0.526** | **0.502** | 0.465 | 0.440 |
| CodeLlama-70B | 0.526 | 0.490 | **0.553** | **0.529** | 0.535 | 0.499 | 0.535 | 0.512 |
| DeepSeek-Coder-33B | 0.474 | 0.444 | **0.518** | **0.480** | 0.491 | 0.463 | 0.465 | 0.439 |
| yang-z/CodeV-QC-7B | 0.526 | 0.474 | 0.526 | 0.472 | **0.544** | **0.488** | 0.509 | 0.472 |
| RTLCoder-DeepSeek-V1 | 0.483 | 0.409 | 0.465 | 0.437 | 0.439 | 0.399 | **0.474** | **0.450** |
| VeriThoughts-Instruct-7B | 0.386 | 0.368 | 0.404 | 0.389 | **0.421** | **0.411** | 0.342 | 0.323 |

sign metrics, enabling the evaluation of LLM-generated code for efficiency across multiple objectives. To generalize the efficiency score to multiple metrics, we consider a task-specific metric set $\mathcal{M} \subseteq \{\text{area, delay, power}\}$, where the size of $\mathcal{M}$ determines the dimensionality of the objective (e.g., two metrics for area–delay optimization or all three for full PPA evaluation). For each metric $m \in \mathcal{M}$, let $e_{i,j}^{(m)}$ denote the single-metric efficiency score (defined in Eq. 1). We then compute a multi-objective efficiency score as a weighted combination defined in Eq. 3. Finally, we obtain the multi-objective eff@$k$ by substituting $e_{i,j}^{\text{multi}}$ for $e_{i,j}$ in Eq. 2.

$$e_{i,j}^{\text{multi}} = \sum_{m \in \mathcal{M}} w_m \, e_{i,j}^{(m)}, \qquad \sum_{m \in \mathcal{M}} w_m = 1, \; w_m \geq 0. \tag{3}$$

Table 4 shows the performance of different LLMs on multi-objective optimization across varying metric combinations. For each metric set $\mathcal{M}$, we report the multi-objective eff@k scores with equal weighting ($w_m{=}1/|\mathcal{M}|$) across all metrics in the set. The results show that, unlike single-metric evaluation, there is no universally easiest metric combination. For example, GPT-4o-mini peaks on Area–Power, while the Llama3.3-70B-Instruct model achieves its highest score on Area–Delay. Several mid-sized models, such as Mixtral and StarCoder2, achieve their best performance on Delay–Power. Across most models, the lowest scores appear on the Area–Delay or Area-Delay-Power combinations, reflecting the inherently competing nature of these metrics. Overall, these results indicate that multi-objective PPA optimization remains a challenging problem, and different LLMs exhibit different strengths depending on the metric set.

## 6 FAILURE ANALYSIS AND INSIGHTS

While LLMs reliably produce functionally correct Verilog, their ability to optimize is uneven across metrics. Area optimization is comparatively tractable, since it often reduces to logic simplification or FSM re-encoding. These area optimizations represent common, syntactic, pattern-like edits that appear frequently in code corpora and programmer-annotated examples, making them more learnable for LLMs. By contrast, delay requires identifying and shortening the critical path, and power depends on subtle factors like switching activity and memory usage. Crucially, area transforms are often local (e.g., simplifying a logic expression), while power and delay optimization typically requires global reasoning across the entire design (e.g., pipeline balancing, critical path restructuring). Current LLMs, especially smaller ones, struggle with these larger-scope transformations. This difficulty is reflected in our quadrant analysis (Figure 5a and 8), where many delay- and power-optimized designs remain correct but fail to improve efficiency, whereas area optimizations succeed more often. Moreover, analysis of optimization strategies (Figure 5b) shows that the hardest transformations are register optimizations for delay, followed by resource sharing for power, and sequential restructuring for delay. In contrast, strategies tied to area are easier, aligning with our observation that area is the most accessible metric for LLMs.

Model scale and specialization strongly influence outcomes. Larger models (15B, 33B, 70B) capture richer patterns and propose alternative architectures, showing stronger generalization across

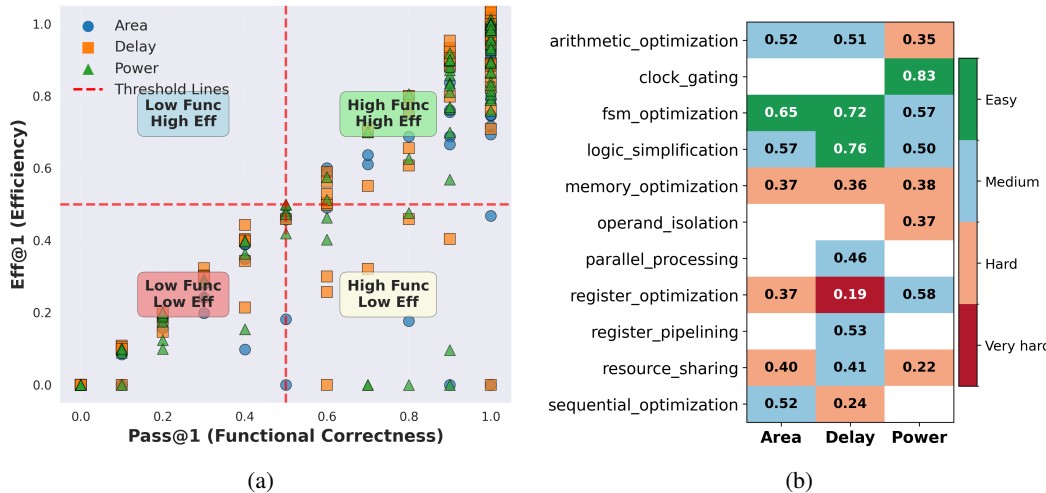

Figure 5: Failure mode analysis of optimization outcomes. (a) Quadrant plot showing the correlation between functional correctness (*Pass@1*) and synthesis efficiency (*Eff@1*) across area, delay, and power objectives. (b) Heatmap of optimization strategy difficulty across different optimization objectives area, delay, and power.

most optimization tactics. For example, larger code models perform more consistently on FSM optimization compared to smaller models, though they still struggle with register optimization and resource sharing. Models with explicit reasoning traces (e.g., DeepSeek, VeriThoughts) better decompose transformations and achieve stronger optimizations. Interestingly, register optimization lags behind other strategies across all DeepSeek variants (DeepSeek-Chat, DeepSeek-Coder, RTLCoder-DeepSeek-V1), suggesting this tactic requires capabilities beyond what current architectures capture, regardless of scale or specialization. Domain-tuned models outperform code models, which in turn outperform general-purpose LLMs, showing the value of Verilog-specific pretraining. Notably, generalist chat models show weaker performance on power-optimization techniques like clock gating and operand isolation, which are less prevalent in general training datasets.

Finally, a fundamental limitation is that Verilog training data lacks efficiency labels. LLMs therefore default to surface-level pattern matching rather than structural reasoning, and without feedback or synthesis-in-the-loop, they cannot tell whether changes reduce gate count or lengthen the critical path. Completion-style models exacerbate this issue, often rephrasing the baseline instead of innovating, whereas instruction-tuned models attempt more substantive edits. These findings suggest that true progress will require efficiency-focused benchmarks such as Pluto to guide future advances.

# 7 CONCLUSION

In this paper, we introduced *Pluto*, a comprehensive benchmark designed to evaluate the synthesis efficiency of LLM-generated Verilog code. *Pluto* provides an evaluation set of 114 hardware design problems, each accompanied by three reference optimized implementations (targeting area, delay, and power), an unoptimized baseline, a self-checking testbench, and a natural language description. Experimental results show that while LLMs can achieve high functional correctness, reaching up to 78.3% at pass@1, their synthesis efficiency remains limited: area efficiency of 63.8%, delay efficiency of 65.9%, and power efficiency of 64.0% at eff@1 compared to expert-crafted designs. These findings highlight the importance of efficiency-aware benchmarks beyond correctness alone and highlights the current limitations of LLMs in hardware optimization.

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

## A  APPENDIX

### A.1  UNOPTIMIZED CODE AND TESTBENCH FOR PROBLEM #17 (TRAILING ZEROS) IN FIGURE 1

#### A.1.1  UNOPTIMIZED CODE

```verilog
module unopt_model #(parameter
  DATA_WIDTH = 32
) (
  input  [DATA_WIDTH-1:0] din,
  output logic [$clog2(DATA_WIDTH):0] dout
);

    logic [DATA_WIDTH-1:0] din_adj;
    logic [$clog2(DATA_WIDTH):0] idx;

    always_comb begin
                 idx = 0;
                 din_adj = din & (~din+1);
                 for (int i=0; i<DATA_WIDTH; i++) begin
                         idx += (din_adj[i]) ? i : 0;
                 end
    end

    assign dout = (din_adj == 0 ? DATA_WIDTH : din_adj == 1 ? 0 : idx);

endmodule
```

#### A.1.2  SELF-CHECKING TESTBENCH

```verilog
`timescale 1 ps/1 ps

module tb();

    reg clk = 0;
    initial forever #5 clk = ~clk;

    wire [5:0] dout_opt, dout_unopt;
    reg [31:0] din;

    integer errors = 0;
    integer errortime = 0;
    integer clocks = 0;
    integer total_cycles = 200;

    initial begin
```

```
17            $dumpfile("wave.vcd");
18            $dumpvars(1, clk, din, dout_opt, dout_unopt);
19
20            // Initialize din to avoid X values
21            din = 0;
22
23            // Generate random values for din
24            repeat(total_cycles) @(posedge clk) din = $random;
25        end
26
27        wire tb_match;
28        assign tb_match = (dout_opt === dout_unopt);
29
30        opt_model opt_model (
31            .din(din),
32            .dout(dout_opt)
33        );
34
35        unopt_model unopt_model (
36            .din(din),
37            .dout(dout_unopt)
38        );
39
40        always @(posedge clk) begin
41            clocks = clocks + 1;
42            if (!tb_match) begin
43                if (errors == 0) errortime = $time;
44                errors = errors + 1;
45            end
46
47            // Print the signals for debugging
48            $display("Time=%0t␣|␣Cycle=%0d␣|␣din=%h␣|␣opt=%h␣|␣unopt=%h␣|␣match=%b",
49                $time, clocks, din, dout_opt, dout_unopt, tb_match);
50
51            if (clocks >= total_cycles) begin
52                $display("Simulation␣completed.");
53                $display("Total␣mismatches:␣%1d␣out␣of␣%1d␣samples", errors, clocks);
54                $display("Simulation␣finished␣at␣%0d␣ps", $time);
55                $finish;
56            end
57        end
58
59        initial begin
60            #1000000
61            $display("TIMEOUT");
62            $finish();
63        end
64
65    endmodule
```

## A.2 OPTIMIZATION STRATEGIES

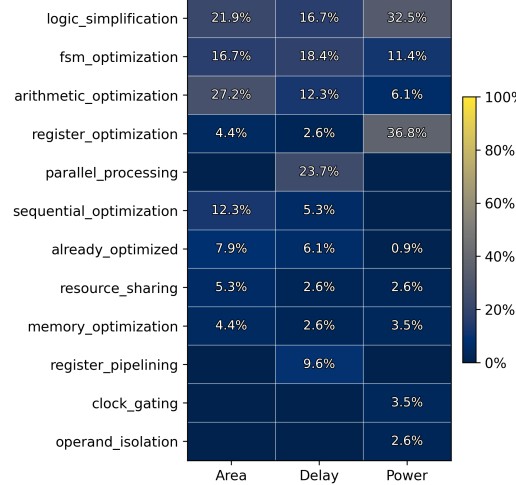

Figure 6: Optimization strategies employed for area, delay, and power improvements.

## A.3 CODE OF EXAMPLE PROBLEMS IN TABLE 2

In the example problems shown in the appendix, the area- and power-optimized solutions coincided, as area-oriented designs also achieved the best power results, and vice versa. This overlap arises because common power-saving techniques, such as clock gating and operand isolation, were not applicable as some designs lacked a clock signal, while others did not include an enable signal. Consequently, explicit power-specific transformations could not be meaningfully applied. Moreover, in certain cases, power optimizations indirectly reduced area, further reinforcing the convergence of the two objectives into a single optimized implementation.

### A.3.1 PROBLEM #60: RTLLM ALU

**Problem description:** Implement a 32-bit Arithmetic Logic Unit (ALU) for a MIPS-ISA CPU. The ALU takes two 32-bit operands (a and b) and a 6-bit control signal (aluc) that specifies which operation to perform. Based on this control signal, the ALU produces a 32-bit result (r) and several status outputs: zero indicates whether the result is zero, carry flags if a carry occurred, negative shows if the result is negative, overflow signals arithmetic overflow, and flag is used for set-less-than instructions (slt and sltu). The module supports arithmetic, logical, shift, and immediate load operations defined by specific opcodes (e.g., ADD, SUB, AND, OR, XOR, SLT, LUI).
**Benchmark solution:** ALU implementation with case statement and parameterized opcodes.

```verilog
module unopt_model(
    input [31:0] a,
    input [31:0] b,
    input [5:0] aluc,
    output [31:0] r,
    output zero,
    output carry,
    output negative,
    output overflow,
    output flag
    );

    parameter ADD = 6'b100000;
    parameter ADDU = 6'b100001;
    parameter SUB = 6'b100010;
    parameter SUBU = 6'b100011;
    parameter AND = 6'b100100;
    parameter OR = 6'b100101;
    parameter XOR = 6'b100110;
    parameter NOR = 6'b100111;
    parameter SLT = 6'b101010;
    parameter SLTU = 6'b101011;
    parameter SLL = 6'b000000;
    parameter SRL = 6'b000010;
    parameter SRA = 6'b000011;
    parameter SLLV = 6'b000100;
    parameter SRLV = 6'b000110;
    parameter SRAV = 6'b000111;
    parameter JR = 6'b001000;
    parameter LUI = 6'b001111;

    wire signed [31:0] a_signed;
    wire signed [31:0] b_signed;

    reg [32:0] res;

    assign a_signed = a;
    assign b_signed = b;
    assign r = res[31:0];

    assign flag = (aluc == SLT || aluc == SLTU) ? ((aluc == SLT) ? (a_signed < b_signed) : (a < b)) : 1'bz;
    assign zero = (res == 32'b0) ? 1'b1 : 1'b0;

    always @ (a or b or aluc)
    begin
        case(aluc)
            ADD: begin
                res <= a_signed + b_signed;
            end
            ADDU: begin
                res <= a + b;
            end
            SUB: begin
```

```verilog
                        res <= a_signed - b_signed;
                    end
                SUBU: begin
                        res <= a - b;
                    end
                AND: begin
                        res <= a & b;
                    end
                OR: begin
                        res <= a | b;
                    end
                XOR: begin
                        res <= a ^ b;
                    end
                NOR: begin
                        res <= ~(a | b);
                    end
                SLT: begin
                        res <= a_signed < b_signed ? 1 : 0;
                    end
                SLTU: begin
                        res <= a < b ? 1 : 0;
                    end
                SLL: begin
                        res <= b << a;
                    end
                SRL: begin
                        res <= b >> a;
                    end
                SRA: begin
                        res <= b_signed >>> a_signed;
                    end
                SLLV: begin
                        res <= b << a[4:0];
                    end
                SRLV: begin
                        res <= b >> a[4:0];
                    end
                SRAV: begin
                        res <= b_signed >>> a_signed[4:0];
                    end
                LUI: begin
                        res <= {a[15:0], 16'h0000};
                    end
                default:
                begin
                        res <= 32'bz;
                    end
            endcase
        end
endmodule
```

**Our expert-written area and power optimized solution:** Shared adder for arithmetic, simplified flag logic and operand reuse, leading to 26% area reduction. Operand gating and early zeroing for large shifts to cut switching activity, for 4% power reduction.

```verilog
    wire sub_mode   = (aluc==SUB) | (aluc==SUBU) | (aluc==SLT) | (aluc==SLTU);
    wire [31:0] b_eff = sub_mode ? ~b : b;
    wire        cin   = sub_mode;
    wire [32:0] sum33 = {1'b0,a} + {1'b0,b_eff} + cin;

    wire [31:0] add_res   = sum33[31:0];
    wire        add_carry = sum33[32];

    wire ovf = (a[31]^add_res[31]) & (b_eff[31]^add_res[31]);

    wire signed_lt  = add_res[31] ^ ovf;
    wire uns_lt     = ~add_carry;
    wire [31:0] slt_res  = {31'b0, signed_lt};
    wire [31:0] sltu_res = {31'b0, uns_lt};

    wire [31:0] and_res = a & b;
    wire [31:0] or_res  = a | b;
    wire [31:0] xor_res = a ^ b;
    wire [31:0] nor_res = ~(a | b);

    wire [4:0] sa5    = a[4:0];
    wire       any_hi = |a[31:5];
```

```verilog
25      wire [31:0] sll_full = any_hi ? 32'b0        : (b <<  sa5);
26      wire [31:0] srl_full = any_hi ? 32'b0        : (b >>  sa5);
27      wire [31:0] sra_full = any_hi ? {32{b[31]}} : ($signed(b) >>> sa5);
28
29      wire [31:0] sllv_res = (b <<  a[4:0]);
30      wire [31:0] srlv_res = (b >>  a[4:0]);
31      wire [31:0] srav_res = ($signed(b) >>> a[4:0]);
32
33      wire [31:0] lui_res = {a[15:0], 16'h0000};
34
35      reg [31:0] r_int;
36      always @* begin : result_mux
37          (* parallel_case, full_case *)
38          case (aluc)
39              ADD, ADDU:  r_int = add_res;
40              SUB, SUBU:  r_int = add_res;
41              AND:        r_int = and_res;
42              OR:         r_int = or_res;
43              XOR:        r_int = xor_res;
44              NOR:        r_int = nor_res;
45              SLT:        r_int = slt_res;
46              SLTU:       r_int = sltu_res;
47              SLL:        r_int = sll_full;
48              SRL:        r_int = srl_full;
49              SRA:        r_int = sra_full;
50              SLLV:       r_int = sllv_res;
51              SRLV:       r_int = srlv_res;
52              SRAV:       r_int = srav_res;
53              LUI:        r_int = lui_res;
54              JR:         r_int = 32'bz;
55              default:    r_int = 32'bz;
56          endcase
57      end
58
59      assign r    = r_int;
60      assign zero = ~(|r_int);
61
62      assign carry    = 1'bz;
63      assign overflow = 1'bz;
64      assign negative = 1'bz;
65      assign flag     = (aluc==SLT)  ? signed_lt :
66                        (aluc==SLTU) ? uns_lt    :
67                                       1'bz;
68  endmodule
```

**Our expert-written delay optimized solution:** Parallel datapaths with one-hot muxing for shallow critical path, leading to 26% delay reduction.

```verilog
1       wire signed [31:0] a_signed = a;
2       wire signed [31:0] b_signed = b;
3
4       wire [31:0] add_u  = a + b;
5       wire [31:0] sub_u  = a - b;
6       wire signed [31:0] add_s = a_signed + b_signed;
7       wire signed [31:0] sub_s = a_signed - b_signed;
8
9       wire [31:0] and_res = a & b;
10      wire [31:0] or_res  = a | b;
11      wire [31:0] xor_res = a ^ b;
12      wire [31:0] nor_res = ~(a | b);
13
14      wire slt_res  = (a_signed < b_signed);
15      wire sltu_res = (a < b);
16
17      wire [4:0] shamt5 = a[4:0];
18
19
20      wire [31:0] sll_full  = (b <<  a);              // full 'a'
21      wire [31:0] srl_full  = (b >>  a);              // full 'a'
22      wire [31:0] sra_full  = ($signed(b) >>> a_signed);// full signed 'a'
23      wire [31:0] sllv_res  = (b <<  shamt5);
24      wire [31:0] srlv_res  = (b >>  shamt5);
25      wire [31:0] srav_res  = ($signed(b) >>> shamt5);
26
27      wire [31:0] lui_res = {a[15:0], 16'h0000};
28
29      wire sel_ADD  = (aluc==ADD);
30      wire sel_ADDU = (aluc==ADDU);
31      wire sel_SUB  = (aluc==SUB);
32      wire sel_SUBU = (aluc==SUBU);
```

```
33        wire sel_AND  = (aluc==AND);
34        wire sel_OR   = (aluc==OR);
35        wire sel_XOR  = (aluc==XOR);
36        wire sel_NOR  = (aluc==NOR);
37        wire sel_SLT  = (aluc==SLT);
38        wire sel_SLTU = (aluc==SLTU);
39        wire sel_SLL  = (aluc==SLL);
40        wire sel_SRL  = (aluc==SRL);
41        wire sel_SRA  = (aluc==SRA);
42        wire sel_SLLV = (aluc==SLLV);
43        wire sel_SRLV = (aluc==SRLV);
44        wire sel_SRAV = (aluc==SRAV);
45        wire sel_LUI  = (aluc==LUI);
46        wire sel_JR   = (aluc==JR);
47
48        wire any_sel = sel_ADD|sel_ADDU|sel_SUB|sel_SUBU|sel_AND|sel_OR|sel_XOR|sel_NOR|
49                       sel_SLT|sel_SLTU|sel_SLL|sel_SRL|sel_SRA|sel_SLLV|sel_SRLV|sel_SRAV|sel_LUI;
50
51        wire [31:0] r_known =
52               (sel_ADD  ? add_s  : 32'b0) |
53               (sel_ADDU ? add_u  : 32'b0) |
54               (sel_SUB  ? sub_s  : 32'b0) |
55               (sel_SUBU ? sub_u  : 32'b0) |
56               (sel_AND  ? and_res: 32'b0) |
57               (sel_OR   ? or_res : 32'b0) |
58               (sel_XOR  ? xor_res: 32'b0) |
59               (sel_NOR  ? nor_res: 32'b0) |
60               (sel_SLT  ? {31'b0, slt_res } : 32'b0) |
61               (sel_SLTU ? {31'b0, sltu_res} : 32'b0) |
62               (sel_SLL  ? sll_full : 32'b0) |
63               (sel_SRL  ? srl_full : 32'b0) |
64               (sel_SRA  ? sra_full : 32'b0) |
65               (sel_SLLV ? sllv_res : 32'b0) |
66               (sel_SRLV ? srlv_res : 32'b0) |
67               (sel_SRAV ? srav_res : 32'b0) |
68               (sel_LUI  ? lui_res  : 32'b0);
69
70        assign r = (any_sel && !sel_JR) ? r_known : 32'bz;
71
72        assign zero = (r == 32'b0) ? 1'b1 : 1'b0;
73
74        assign flag = (sel_SLT)  ? slt_res  :
75                      (sel_SLTU) ? sltu_res :
76                                   1'bz;
77
78        assign carry    = 1'bz;
79        assign negative = 1'bz;
80        assign overflow = 1'bz;
81   endmodule
```

### A.3.2  PROBLEM #68: RTLLM FSM

**Problem description:** Implement a finit state machine (FSM) that detects the input sequence 10011 on a single-bit input stream. The module has three inputs: the serial input bit (IN), the clock (CLK), and a synchronous reset (RST). It produces one output, MATCH, which is asserted high when the specified sequence is recognized. The FSM supports continuous input and loop detection. When reset is active, the FSM initializes and MATCH is cleared to 0. The output MATCH is asserted during the cycle when the last 1 of the target sequence is received, and the design ensures that repeated or overlapping patterns (e.g., 100110011) correctly generate multiple match pulses.

**Benchmark solution:** States are binary-encoded, with sequential next-state logic in a Mealy FSM while output occupies a register.

```
1    module unopt_model (
2        input  wire IN,
3        input  wire CLK,
4        input  wire RST,
5        output wire MATCH
6    );
7
8    reg [2:0] ST_cr,ST_nt;
9
10   parameter s0 = 3'b000;
11   parameter s1 = 3'b001;
12   parameter s2 = 3'b010;
13   parameter s3 = 3'b011;
14   parameter s4 = 3'b100;
```

```verilog
parameter s5 = 3'b101;

always@(posedge CLK or posedge RST) begin
     if(RST)
          ST_cr <= s0;
     else
          ST_cr <= ST_nt;
end

always@(*) begin
     case(ST_cr)
          s0:begin
               if (IN==0)
                    ST_nt = s0;
               else
                    ST_nt = s1;
          end

          s1:begin
                    if (IN==0)
                         ST_nt = s2;
                    else
                         ST_nt = s1;
               end

          s2:begin
                    if (IN==0)
                         ST_nt = s3;
                    else
                         ST_nt = s1;
               end

          s3:begin
                    if (IN==0)
                         ST_nt = s0;
                    else
                         ST_nt = s4;
               end

          s4:begin
                    if (IN==0)
                         ST_nt = s2;
                    else
                         ST_nt = s5;
               end

          s5:begin
                    if (IN==0)
                         ST_nt = s2;
                    else
                         ST_nt = s1;
               end

     endcase
end

always@(*) begin
          if(RST)
               MATCH  <= 0;
          else if (ST_cr == s4 && IN == 1)
               MATCH  <= 1;
          else
               MATCH  <= 0;
end

endmodule
```

**Our expert-written area and power optimized solution:** Casez-based transitions and direct Mealy output computation, removing extra register, leading to 32% area reduction. Compact binary encoding and casez-based transitions to cut toggling for 23% power reduction.

```verilog
localparam [2:0] s0=3'b000, s1=3'b001, s2=3'b010,
                 s3=3'b011, s4=3'b100, s5=3'b101;

reg [2:0] ST_cr, ST_nt;

always @(posedge CLK or posedge RST) begin
     if (RST)
          ST_cr <= s0;
     else
```

```
10          ST_cr <= ST_nt;
11      end
12
13      always @* begin
14          ST_nt = s0;
15          casez ({ST_cr, IN})
16              // s0: 0->s0, 1->s1
17              {s0,1'b0}: ST_nt = s0;
18              {s0,1'b1}: ST_nt = s1;
19
20              // s1: 0->s2, 1->s1
21              {s1,1'b0}: ST_nt = s2;
22              {s1,1'b1}: ST_nt = s1;
23
24              // s2: 0->s3, 1->s1
25              {s2,1'b0}: ST_nt = s3;
26              {s2,1'b1}: ST_nt = s1;
27
28              // s3: 0->s0, 1->s4
29              {s3,1'b0}: ST_nt = s0;
30              {s3,1'b1}: ST_nt = s4;
31
32              // s4: 0->s2, 1->s5
33              {s4,1'b0}: ST_nt = s2;
34              {s4,1'b1}: ST_nt = s5;
35
36              // s5: 0->s2, 1->s1
37              {s5,1'b0}: ST_nt = s2;
38              {s5,1'b1}: ST_nt = s1;
39
40              default:   ST_nt = s0;
41          endcase
42      end
43
44      assign MATCH = (ST_cr == s4) & IN;
45
46  endmodule
```

**Our expert-written delay optimized solution:** One-hot state encoding with pre-decoded inputs and Moore-style output, leading to 23% delay reduction.

```
1
2      reg [5:0] S, S_next;
3
4      reg [5:0] S, S_next;
5
6      always @(posedge CLK or posedge RST) begin
7          if (RST)
8              S <= 6'b000001;          // s0
9          else
10             S <= S_next;
11     end
12
13     wire in1 =  IN;
14     wire in0 = ~IN;
15
16     always @* begin
17         S_next[0] = (S[0] & in0) | (S[3] & in0);              // -> s0
18         S_next[1] = (S[0] & in1) | (S[1] & in1) | (S[2] & in1)
19                   | (S[5] & in1);                             // -> s1
20         S_next[2] = (S[1] & in0) | (S[4] & in0) | (S[5] & in0);  // -> s2
21         S_next[3] = (S[2] & in0);                             // -> s3
22         S_next[4] = (S[3] & in1);                             // -> s4
23         S_next[5] = (S[4] & in1);                             // -> s5
24     end
25
26     always @(posedge CLK or posedge RST) begin
27         if (RST)
28             MATCH <= 1'b0;
29         else
30             MATCH <= S[5];
31     end
32 endmodule
```

### A.3.3   PROBLEM #87: VERILOGEVAL PROB095

**Problem description:** Implement a module that generates a control signal (shift_ena) for a shift register. The module has a clock input (clk), a synchronous active-high reset (reset), and a single

output (shift_ena). The functionality requires that when the FSM is reset, the shift_ena signal is asserted high for exactly four consecutive clock cycles before being deasserted permanently. After this sequence, shift_ena remains low indefinitely until another reset occurs, at which point the behavior repeats. All sequential operations are triggered on the positive edge of the clock.

**Benchmark solution:** Uses explicit states with next-state logic to drive shift_ena.

```verilog
module unopt_model (
  input  clk,
  input  reset,
  output reg shift_ena
);
  parameter B0=0, B1=1, B2=2, B3=3, Done=4;

  reg [2:0] state, next;

  always @* begin
    case (state)
      B0:   next = B1;
      B1:   next = B2;
      B2:   next = B3;
      B3:   next = Done;
      Done: next = Done;
      default: next = B0;
    endcase
  end

  always @(posedge clk) begin
    if (reset) begin
      state    <= B0;
      shift_ena <= 1'b1;
    end else begin
      state    <= next;
      shift_ena <= (next != Done);
    end
  end
endmodule
```

**Our expert-written area and power optimized solution:** Minimized register width, using a 2-bit counter, and compact comparator logic, leading to 47% area reduction. Small counter reused which reduced toggling activity to minimize dynamic power, for 46% power reduction.

```verilog
reg [1:0] counter; // 2 bits are enough to count up to 4

always @(posedge clk) begin
  if (reset) begin
    counter <= 2'b00; // Reset counter
    shift_ena <= 1'b1; // Enable on reset
  end else if (counter < 2'b11) begin
    counter <= counter + 1; // Increment counter
    shift_ena <= 1'b1; // Keep shift_ena high while counting
  end else begin
    shift_ena <= 1'b0; // Disable after 4 cycles
  end
end

endmodule
```

**Our expert-written delay optimized solution:** Wider counter, using 3 bits, to simplify comparison and reduce logic depth on the critical path, leading to 37% delay reduction.

```verilog
reg [2:0] count; // 3-bit counter to count 4 cycles

always @(posedge clk) begin
  if (reset) begin
    count <= 3'b000; // Reset count
    shift_ena <= 1'b1; // Enable shift initially
  end else if (count < 3'b011) begin
    count <= count + 1; // Increment count
    shift_ena <= 1'b1; // Keep shift enabled
  end else begin
    shift_ena <= 1'b0; // Disable shift after 4 cycles
  end
end

endmodule
```

### A.3.4 PROBLEM #104: VERILOGEVAL PROB136

**Problem description:** Implement a cellular automaton game, similar to Conway's Game of Life, on a 16x16 grid. The grid is represented as a 256-bit vector (q), where each row of 16 cells maps to a sub-vector, and each cell can be alive (1) or dead (0). The module has a clock input (clk), a load signal (load) for synchronously loading an initial 256-bit state (data) into q, and produces the updated 256-bit grid state as output. At every positive clock edge, the grid advances by one timestep, with each cell's next state determined by its number of neighbors: cells die with fewer than 2 or more than 3 neighbors, remain unchanged with exactly 2 neighbors, and become alive with exactly 3 neighbors. The grid is modeled as a toroid, meaning edges wrap around so that cells on the boundaries consider neighbors from the opposite side.

**Benchmark solution:** Straightforward RTL with per-cell neighbor recomputation and sequential summing.

```verilog
module unopt_model (
  input clk,
  input load,
  input [255:0] data,
  output reg [255:0] q
);

  logic [323:0] q_pad;
  always@(*) begin
    for (int i=0;i<16;i++)
      q_pad[18*(i+1)+1 +: 16] = q[16*i +: 16];
    q_pad[1 +: 16] = q[16*15 +: 16];
    q_pad[18*17+1 +: 16] = q[0 +: 16];

    for (int i=0; i<18; i++) begin
      q_pad[i*18] = q_pad[i*18+16];
      q_pad[i*18+17] = q_pad[i*18+1];
    end
  end

  always @(posedge clk) begin
    for (int i=0;i<16;i++)
    for (int j=0;j<16;j++) begin
      q[i*16+j] <=
        ((q_pad[(i+1)*18+j+1 -1+18] + q_pad[(i+1)*18+j+1 +18] + q_pad[(i+1)*18+j+1 +1+18] +
        q_pad[(i+1)*18+j+1 -1]                              + q_pad[(i+1)*18+j+1+1] +
        q_pad[(i+1)*18+j+1 -1-18]   + q_pad[(i+1)*18+j+1 -18] + q_pad[(i+1)*18+j+1 +1-18]) & 3'h7 | q[i*16+j]) == 3'h3;
    end

    if (load)
      q <= data;

  end

endmodule
```

**Our expert-written area and power optimized solution:** Sharing per-row horizontal sums and bitwise rotations for toroidal wrap, minimizing summations for 19% area reduction. Computation reuse decreasing toggling, leading to 36% power reduction.

```verilog
// --- Helpers ----------------------------------------------------------
function automatic [7:0] idx(input [3:0] r, input [3:0] c);
  idx = {r, c}; // r*16 + c
endfunction

// Bit-rotate wires (toroidal wrap) - wiring only (no logic area)
function automatic [15:0] rol1(input [15:0] x); rol1 = {x[14:0], x[15]}; endfunction
function automatic [15:0] ror1(input [15:0] x); ror1 = {x[0],    x[15:1]}; endfunction

// Add-three 1-bit vectors in parallel: returns {carry,sum}
// a+b+c = sum ^ (2*carry) per bit
function automatic [31:0] add3_vec(input [15:0] a, input [15:0] b, input [15:0] c);
  add3_vec[15:0]  = a ^ b ^ c;                    // sum (LSB)
  add3_vec[31:16] = (a & b) | (a & c) | (b & c);  // carry (means +2)
endfunction

// --- Unpack rows (wires) ----------------------------------------------
wire [15:0] row [15:0];
genvar ur;
generate
  for (ur = 0; ur < 16; ur = ur + 1) begin : UNPACK
    assign row[ur] = q[{ur[3:0], 4'b0000} +: 16];
```

```
 23       end
 24     endgenerate
 25
 26     // --- Precompute per-row horizontal neighbors (shared) ----------------------
 27     wire [15:0] rol [15:0], ror [15:0];
 28     wire [15:0] sTrip [15:0], cTrip [15:0];  // for (L,C,R) of each row (0..3)
 29     wire [15:0] sPair [15:0], cPair [15:0];  // for (L,R) of each row (0..2)
 30
 31     genvar hr;
 32     generate
 33       for (hr = 0; hr < 16; hr = hr + 1) begin : HROW
 34         assign rol[hr] = rol1(row[hr]);
 35         assign ror[hr] = ror1(row[hr]);
 36
 37         // Triplet = left + center + right (encoded as s + 2*c)
 38         wire [31:0] trip_pack = add3_vec(rol[hr], row[hr], ror[hr]);
 39         assign sTrip[hr] = trip_pack[15:0];
 40         assign cTrip[hr] = trip_pack[31:16];
 41
 42         // Pair = left + right
 43         assign sPair[hr] = rol[hr] ^ ror[hr];
 44         assign cPair[hr] = rol[hr] & ror[hr];
 45       end
 46     endgenerate
 47
 48     integer r;
 49     reg [255:0] nxt;
 50
 51     reg [3:0]  rn, rp;
 52     reg [15:0] sT, cT, sM, cM, sB, cB;
 53     reg [15:0] sS, cS;                    // sum of (sT + sM + sB) as s + 2*c
 54     reg [15:0] U_is0, U_ge2, U_is1;    // onehot(U) for U = cS + cT + cM + cB
 55     reg [15:0] is3, is2;
 56
 57     always @* begin
 58       nxt = '0;
 59
 60       for (r = 0; r < 16; r = r + 1) begin
 61         rn = (r == 0 ) ? 4'd15 : r - 1;
 62         rp = (r == 15) ? 4'd0  : r + 1;
 63
 64         sT = sTrip[rn]; cT = cTrip[rn];    // top triplet from row r-1
 65         sM = sPair[r ]; cM = cPair[r ];    // middle pair (no center) from row r
 66         sB = sTrip[rp]; cB = cTrip[rp];    // bottom triplet from row r+1
 67
 68         {cS, sS} = add3_vec(sT, sM, sB);
 69
 70         U_is0 = ~(cS | cT | cM | cB);
 71         U_ge2 = ( (cS & cT) | (cS & cM) | (cS & cB)
 72                 | (cT & cM) | (cT & cB) | (cM & cB) );
 73         U_is1 = ~(U_is0 | U_ge2);
 74
 75         is3 =  sS & U_is1;
 76         is2 = ~sS & U_is1;
 77
 78         nxt[{r[3:0], 4'b0000} +: 16] = is3 | (row[r] & is2);
 79       end
 80     end
 81
 82     always @(posedge clk) begin
 83       if (load)
 84         q <= data;
 85       else
 86         q <= nxt;
 87     end
 88   endmodule
```

**Our expert-written delay optimized solution:** Parallel neighbor summation with carry-save adder tree and direct decode for 2 and 3, for shallow critical path, leading to 37% delay reduction.

```
 1     function automatic [7:0] idx(input [3:0] r, input [3:0] c);
 2       idx = {r, c};
 3     endfunction
 4
 5     function automatic [15:0] rol1(input [15:0] x); rol1 = {x[14:0], x[15]}; endfunction
 6     function automatic [15:0] ror1(input [15:0] x); ror1 = {x[0],    x[15:1]}; endfunction
 7
 8     function automatic [31:0] add3_vec(input [15:0] a, input [15:0] b, input [15:0] c);
 9       add3_vec[15:0]  = a ^ b ^ c;                      // s
10       add3_vec[31:16] = (a & b) | (a & c) | (b & c);    // c (>=2)
```

```
11    endfunction
12
13    integer r;
14    reg [255:0] nxt;
15
16    reg [3:0] rn, rp;
17    reg [15:0] ru, r0, rd;
18    reg [15:0] ru_l, ru_c, ru_r;
19    reg [15:0] r0_l,        r0_r;
20    reg [15:0] rd_l, rd_c, rd_r;
21
22    reg [15:0] sT, cT, sM, cM, sB, cB, sS, cS;
23    reg [15:0] U_is0, U_ge2, U_is1;  // onehot decode for U = cS+cT+cM+cB
24    reg [15:0] is3, is2;             // neighbor count ==3 / ==2
25
26    always @* begin
27      nxt = '0;
28
29      for (r = 0; r < 16; r = r + 1) begin
30        rn = (r == 0 ) ? 4'd15 : r - 1;
31        rp = (r == 15) ? 4'd0  : r + 1;
32
33        ru = q[{rn,4'b0000} +: 16];
34        r0 = q[{r ,4'b0000} +: 16];
35        rd = q[{rp,4'b0000} +: 16];
36
37        ru_l = rol1(ru);  ru_c = ru;       ru_r = ror1(ru);
38                          r0_l = rol1(r0);      r0_r = ror1(r0);
39        rd_l = rol1(rd);  rd_c = rd;       rd_r = ror1(rd);
40
41        {cT, sT} = add3_vec(ru_l, ru_c, ru_r);   // counts 0..3
42        sM       = (r0_l ^ r0_r);                // pair: 0..2
43        cM       = (r0_l & r0_r);
44        {cB, sB} = add3_vec(rd_l, rd_c, rd_r);
45
46        {cS, sS} = add3_vec(sT, sM, sB);
47
48        U_is0 = ~(cS | cT | cM | cB);
49        U_ge2 = ( (cS & cT) | (cS & cM) | (cS & cB)
50                | (cT & cM) | (cT & cB) | (cM & cB) );
51        U_is1 = ~(U_is0 | U_ge2);
52
53        is3 =  sS & U_is1;
54        is2 = ~sS & U_is1;
55
56        nxt[{r[3:0],4'b0000} +: 16] = is3 | (r0 & is2);
57      end
58    end
59
60    always @(posedge clk) begin
61      if (load)
62        q <= data;
63      else
64        q <= nxt;
65    end
66  endmodule
```

## A.4 PASS@K DEFINITION

The pass@k metric, defined in equation 4, is used for measuring the functional correctness of LLM-generated code. Here, $N$ is the total number of problems in the evaluation set. For a given problem i, we generate $n_i$ samples and evaluate their functional correctness, obtaining $c_i$ correct samples. The metric estimates the probability that at least one of the correct solution appears when drawing $k$ samples.

$$\text{pass@}k = \mathbb{E}_{i=1}^{N}\left[ 1 - \frac{C(n_i - c_i,\, k)}{C(n_i,\, k)} \right] \tag{4}$$

where $C(n,\, k)$ denotes the binomial coefficient 'n choose k' and the term $1 - \frac{C(n_i - c_i,\, k)}{C(n_i,\, k)}$ represents the probability that at least one of the $k$ samples is correct from the $n_i$ total samples.

The metric averages this probability across all $N$ problems to provide an unbiased estimate of the model's functional correctness when generating $k$ samples per problem.

Implementation Note: Following standard practice, we generate $n \geq k$ samples per problem (typically $n = 10$) and compute pass@k for various values of k (e.g., $k \in 1, 5, 10$) to obtain unbiased estimates without requiring k separate evaluation runs.

## A.5 SAMPLING DIVERSITY

In Figure 7, we show three LLM-generated Verilog modules of a parallel-in-serial-out shift register, targeting area optimization, with normalized area efficiency values of 0.924, 0.963 and 1.0, respectively. This diversity demonstrates why eff@k with $k > 1$ is valuable: generating multiple samples increases the likelihood of obtaining high-quality implementations. The optimal sample (rightmost) avoids unnecessary counters and state tracking present in the other solutions, reducing both area and complexity, a correct solution that might not appear in a single generation attempt but becomes accessible when drawing multiple samples.

(a) Top generation at $k = 1$     (b) Top generation at $k = 5$     (c) Top generation at $k = 10$

Figure 7: Three area-optimized implementations of the `piso_shift_register` module (Problem #16) generated at $k \in \{1, 5, 10\}$. The circuit shifts the least significant bit of a multi-bit input `din` to the single-bit output `dout` sequentially, starting when `din_en` goes high. All designs are functionally correct but structurally diverse.

## A.6 FAILURE ANALYSIS

In Figure. 8, we visualize the set of problems that have high pass@k and low eff@k to get better insight on the set of problems that are hard to optimize.

## A.7 PROMPTING WITH OPTIMIZATION STRATEGIES

To assess whether the relatively lower eff@k scores stem from missing optimization guidance, we repeated the experiment using the first prompting strategy, where the LLM rewrites an unoptimized Verilog module into a more efficient implementation, but this time we explicitly provided the optimization goals and permissible strategies for each metric. The results, summarized in Table 5, show that explicit guidance yields only marginal and model-dependent improvements. A few models (e.g., Mixtral, StarCoder2-15B, CodeLlama-70B, and CodeV-QC-7B) show moderate gains across several efficiency metrics, while others improve only in isolated cases (e.g., GPT-4o-mini improves mainly in power, Qwen2.5-Coder-7B only in delay). However, for several strong models, including DeepSeek-Chat and Llama-3.3-70B, efficiency scores worsen despite being given optimization strategies. Overall, the gap between pass@k and eff@k remains substantial, indicating that the limited efficiency performance is not primarily caused by insufficient prompt specification, but rather reflects the underlying difficulty of generating functionally correct and resource-efficient RTL simultaneously.

Table 5: Efficiency results when using the first prompting strategy with explicit optimization goals and metric-specific optimization strategies. While a few models exhibit modest or selective improvements, most models show limited or no gains, and several regress. Overall, providing optimization guidance does not substantially narrow the gap between pass@k and eff@k.

| Model | pass@1 | pass@5 | pass@10 | eff@1 | | | eff@5 | | | eff@10 | | |
|---|---|---|---|---|---|---|---|---|---|---|---|---|
| | | | | Area | Delay | Power | Area | Delay | Power | Area | Delay | Power |
| GPT-4o-mini | 0.471 | 0.667 | 0.725 | 0.435 | 0.440 | 0.423 | 0.622 | 0.637 | 0.642 | 0.696 | 0.693 | 0.716 |
| DeepSeek-Chat | 0.502 | 0.728 | 0.795 | 0.487 | 0.496 | 0.485 | 0.717 | 0.719 | 0.709 | 0.795 | 0.789 | 0.774 |
| Llama-3.3-70B-Instruct | 0.429 | 0.666 | 0.752 | 0.400 | 0.412 | 0.404 | 0.626 | 0.666 | 0.627 | 0.703 | 0.753 | 0.714 |
| Mixtral-8x7B-v0.1 | 0.271 | 0.562 | 0.696 | 0.232 | 0.255 | 0.244 | 0.491 | 0.528 | 0.508 | 0.607 | 0.648 | 0.632 |
| starcoder2-15b-instruct-v0.1 | 0.686 | 0.969 | 0.985 | 0.632 | 0.653 | 0.615 | 0.893 | 0.917 | 0.888 | 0.904 | 0.935 | 0.904 |
| CodeLlama-70b-Instruct-hf | 0.599 | 0.904 | 0.944 | 0.538 | 0.559 | 0.547 | 0.834 | 0.857 | 0.846 | 0.877 | 0.905 | 0.895 |
| DeepSeek-Coder-33B | 0.6088 | 0.9358 | 0.9766 | 0.5539 | 0.5706 | 0.5648 | 0.8745 | 0.8948 | 0.887 | 0.9270 | 0.9514 | 0.9191 |
| Qwen2.5-Coder-7B-Inst | 0.443 | 0.756 | 0.839 | 0.399 | 0.421 | 0.389 | 0.675 | 0.735 | 0.692 | 0.745 | 0.834 | 0.785 |
| yang-z/CodeV-QC-7B | 0.396 | 0.662 | 0.725 | 0.360 | 0.359 | 0.347 | 0.593 | 0.638 | 0.601 | 0.644 | 0.718 | 0.667 |
| RTLCoder-DeepSeek-V1 | 0.470 | 0.816 | 0.895 | 0.418 | 0.428 | 0.413 | 0.738 | 0.780 | 0.726 | 0.828 | 0.860 | 0.803 |
| VeriThoughts-Inst.-7B | 0.545 | 0.770 | 0.836 | 0.475 | 0.508 | 0.492 | 0.673 | 0.715 | 0.699 | 0.748 | 0.774 | 0.758 |

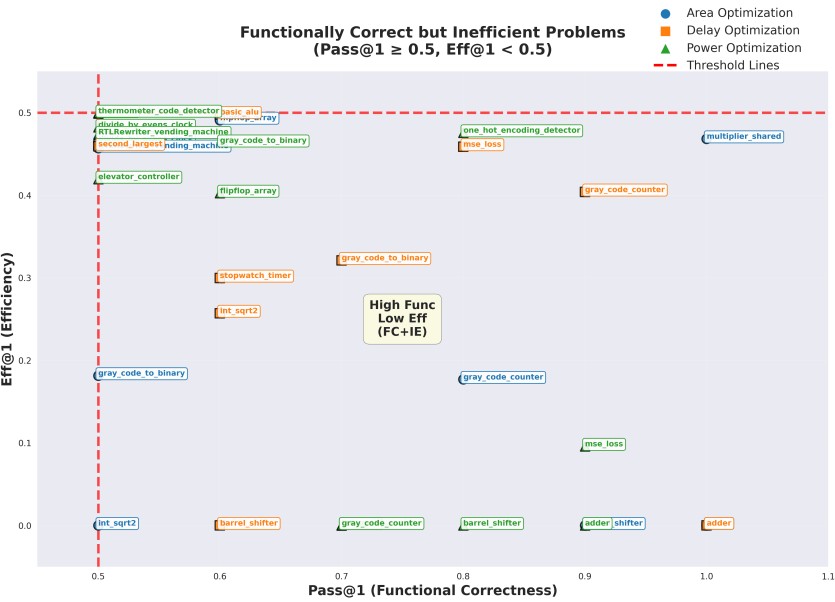

Figure 8: Quadrant plot showing the correlation between functional correctness (*Pass@1*) and synthesis efficiency (*Eff@1*) across area, delay, and power objectives.

