# OpenReview forum: "Pluto: A Benchmark for Evaluating Efficiency of LLM-generated Hardware Code"
_ICLR.cc/2026/Conference — Submitted to ICLR 2026_

### Official Review · Reviewer_jfoZ · 2025-10-28

**Soundness:** 2
**Presentation:** 3
**Contribution:** 2
**Rating:** 2
**Confidence:** 5

**Summary:**

Existing benchmarks for LLM-generated hardware designs primarily focus on functional correctness, neglecting crucial synthesis metrics like area, delay, and power. To address this gap, this work introduces a new benchmark Pluto providing 114 problems with test benches and optimized reference designs.

**Strengths:**

1) This work provides a benchmark with 114 problems, each featuring individually optimized reference designs for area, delay, and power. This establishes a true Pareto front for comprehensive efficiency comparison.

2) This work provides test benches that automatically adapt to different latency requirements and optimization goals, enabling a robust and fair evaluation of designs under various efficiency constraints.

**Weaknesses:**

1) This benchmark relies entirely on expert-designed golden truths and test benches, making it difficult to expand for large-scale assessment.

2) The fundamental need for manually crafted golden implementations is not convincing. It is proposed that predicting Pareto-optimal metrics could potentially eliminate the high cost of expert design and serve the benchmark's evaluation purpose.

3) The benchmark's collection of 114 problems from existing sources (ChipDev, VerilogEval, etc.) is questioned. The core concerns are a lack of justification for its superiority over other benchmarks and an unclear curation strategy, which appears potentially random rather than methodically designed.

4) It is suggested that the lower eff@k scores compared to pass@k might be partly due to a lack of explicit optimization goals and strategies in the prompts, rather than solely the LLMs' inability.

**Questions:**

See the weaknesses.

---

> ### Author Response · Authors · 2025-12-03
> **Response**
>
> We thank the reviewer for their detailed feedback and for crediting Pluto’s strengths in providing “comprehensive efficiency comparison” and “robust and fair evaluation of designs”. We address each of the following concerns as follows:
>
> 1. **Why Expert-Crafted Solutions Are Needed.** We acknowledge the reviewer's observation that Pluto contains expert-designed golden implementations and validated testbenches.This design choice was intentional to ensure benchmarking fidelity. Without verified ground truth references and testbenches, it becomes impossible to determine whether a design is functionally correct or how far is it from the true optimized solution. Expert involvement is required to ensure that the reference implementations both preserve full functionality and represent true Pareto-optimal tradeoffs, providing reliable and interpretable baselines for evaluation.
>
> 2. **Why do we need manually crafted golden implementations v.s a predicted Pareto-optimal metrics?** We agree with the reviewer suggestion that relative comparison between LLM-generated designs could serve as an alternative to golden baselines. However, with this approach we will be blind to how far is LLM generated solution from the true pareto-front. Relative scoring can show which generated solution is better within the produced set, but it cannot answer *how close the solution is to the true achievable optimum for that task*. Without a known reference frontier, two poor designs may still appear optimal relative to each other. Therefore, the golden implementations are needed to provide a true anchor for the PPA efficiency of a particular task.
>
> 3. **Superiority over other benchmarks and Curation Strategy**. Pluto’s superiority over other benchmarks can be summarized as follows:
>     - Pluto is the first benchmark in evaluating LLM-generated hardware code. Unlike previous hardware code benchmarks (like    VerilogEval and RTLLM), Pluto focuses on both functional correctness and efficiency.
>     - Pluto provides three independently optimized canonical implementations per task, each representing an expert-tuned solution for area, delay, or power. While Pluto contains problem statements from both VerilogEval and RTLLM, our canonical solutions are different and much more optimal. As mentioned in the paper, **Pluto’s expert-crafted solutions achieve average improvements of 18.75% (area), 22.75% (delay), and 20.43% (power) over the original RTLLM implementations, and 10.46% (area), 10.33% (delay), and 13.61% (power) over the original VerilogEval designs**. Additionally, Table 2 in the paper provides examples of such improvements.
>     - Pluto is the only code generation benchmark that provides three distinct implementations for different optimization targets. Unlike prior software and hardware coding benchmarks, we are the only benchmark paper that provides Per-Metric optimization.
>
>    **Curation Strategy**. We appreciate the reviewer’s concern and clarify that Pluto’s problem selection process was not arbitrary. The goal of Pluto is to benchmark both functional correctness and hardware efficiency generation, capabilities that cannot be meaningfully evaluated with trivial problems. Therefore, we followed a three-stage curation pipeline:
>     - **Initial Sourcing.** We began by collecting candidate tasks from multiple existing datasets (e.g., VerilogEval, ChipDev, RTLLM) to ensure diversity in problems rather than starting from scratch.
>     - **Filtering for Optimization Potential.** Many problems in earlier datasets are structurally simple and produce identical synthesis results regardless of coding style, leaving no room for optimization tradeoffs. Such tasks were excluded because they cannot meaningfully evaluate efficiency-aware code generation.
>    - **Pareto-Front Validation.** For the remaining problems, we ensured that the three implementations deliver expected superiority in their optimization target under different synthesis strategies (as illustrated in Figure 4 in the paper). Problems that failed to deliver expected tradeoffs were excluded.
>
>    This resulted in 114 tasks in total, each paired with independently optimized implementations for area, delay, and power.
>
> 4. **Prompting with optimization goals and strategies.** We clarify that all results reported in the paper were generated using prompts that explicitly instructed the LLMs to optimize the design for a specific objective (area, delay, or power). In other words, the lower eff@k scores are not a result of missing or vague optimization intent in the prompt rather than true LLM inability to generate optimized code. This was further validated by additional experiments (Appendix A.7, Table 5) where LLM prompt was also augmented with explicit optimization strategies such as register pipelining for delay optimization or resource sharing for area optimization.

---

### Official Review · Reviewer_MZkp · 2025-10-31

**Soundness:** 3
**Presentation:** 3
**Contribution:** 3
**Rating:** 6
**Confidence:** 4

**Summary:**

This paper introduces Pluto, a new benchmark and evaluation framework focused on assessing both the functional correctness and synthesis efficiency (area, delay, power) of Verilog code generated by large language models (LLMs). Pluto comprises 114 hardware design problems, each accompanied by three hand-optimized, Pareto reference implementations targeting area, delay, and power, as well as self-checking, latency-agnostic testbenches for fair comparison. The authors show through extensive experiments that while LLMs can achieve high pass rates for correctness, their designs remain substantially less efficient compared to expert-optimized solutions.

**Strengths:**

1. Pluto addresses clear gaps in existing hardware code benchmarks by providing multiple, per-metric (area, delay, power) Pareto-optimized solutions as ground-truth references for each problem;
2. Problems span a range of difficulty based on established educational and open-source resources, with designs and testbenches developed by independent experts, aiming to minimize contamination and maximize relevance.

**Weaknesses:**

1. While Figure 5 and 8 visualize failure modes, the paper could benefit from a richer exploration of why current LLMs fail on certain optimization tactics (beyond just metric categorization). For example, do larger models show specific structural generalization, or are there systematic syntactic or semantic limitations (e.g., loop unrolling, FSM encoding, arithmetic fusion) that impede efficiency optimization?
2. Inadequate discussion and analysis of key results: Figure 2, which highlights reductions in area, delay, and power, is pivotal to the paper's claims but receives insufficient elaboration in the main text, leaving uncertainty about whether these improvements are representative of the entire dataset or merely selected examples. Similarly, Table 2 is limited to a small number of case studies, lacking comprehensive statistical measures such as variance or standard deviation across all tasks, which undermines the robustness of the findings and could benefit from a more thorough quantitative evaluation.

**Questions:**

How exactly are upper bounds  $T_{i,j}$ in Eq. 1, Page 6 decided? Is there an objective method, or is it ad hoc?

---

> ### Author Response · Authors · 2025-12-03
> **Response**
>
> We thank the reviewer for the detailed and thoughtful evaluation. We are encouraged by the positive assessment of Pluto’s contribution. We address each of the following concerns as follows:
>
> 1. **Failure Exploration.** We thank the reviewer for this valuable suggestion. We have revised Section 6 to provide deeper insights into why LLMs fail at specific optimization tactics, beyond categorical analysis.
>
> 2. **Figure 2 and Table 2 Clarifications.** We have clarified that Figure 2 reports area, delay, and power reductions across all designs in Pluto and expanded the accompanying analysis section accordingly. Table 2 was intended to spotlight a number of case studies from both VerilogEval and RTLLM to showcase that their canonical solutions don’t provide the most efficient implementation. In response to the reviewer’s suggestion, we have included the standard deviations for all averages reported in the papers.
>
> We clarify the raised question below:
>
> **How exactly are upper bounds $T_{i,j}$ in Eq. 1, Page 6 decided? Is there an objective method, or is it ad hoc?**
>
> The upper bound $T_{i,j}$ is not chosen ad-hoc; it is derived directly from optimized reference value $R_{i,j}$. Specifically, we set $T_{i,j} = \alpha R_{i,j}$.  Where $\alpha$ is greater than one and is a fixed constant. This makes the upper bound proportional to the best-known implementation, ensuring that the scoring range [$R_{i,j}$,$T_{i,j}$] scales appropriately with the difficulty and magnitude of each metric. Intuitively, $T_{i,j}$ defines the maximum allowable degradation relative to the optimized baseline. A high $T_{i,j}$ value accommodates a wider range of acceptable designs to receive partial efficiency credits without immediately scoring them as 0 efficiency. We also note that $T_{i,j}$​ is a configurable parameter in our framework, allowing users to adjust the tolerance for degradation depending on their design goals.

---

### Official Review · Reviewer_uhb7 · 2025-11-01

**Soundness:** 3
**Presentation:** 3
**Contribution:** 2
**Rating:** 2
**Confidence:** 4

**Summary:**

This paper introduces Pluto, a benchmark for evaluating both functional correctness and synthesis efficiency (area, delay, and power) of LLM-generated Verilog code. It provides 114 design problems, each with three Pareto-optimal reference implementations and self-checking testbenches. Experiments across various LLMs show that the efficiency of existing models' design still lags far behind expert-designed implementations.

**Strengths:**

1. Pluto fills a missing piece in the LLM-powered EDA field by shifting hardware code evaluation from functional correctness to efficiency.

2. It provides a comprehensive dataset of 114 problems, each with per-metric optimized reference implementations and self-checking testbenches, offering a complete view of LLM-generated Verilog code evaluation.

3. The study across various LLMs shows that existing models still fall short of human experts in producing highly optimized Verilog code, revealing the current limitations of LLMs.

**Weaknesses:**

1. The major concern is that this work provides very limited insights for the ICLR community. While it demonstrates how human experts can craft high-quality Verilog data, it does not offer new or generalizable understanding of how to leverage or improve LLMs, e.g., through LLM-assisted data generation or methods to enhance domain-specific generation quality. The dataset contribution would be more suitable for venues such as DAC or ICCAD.

2. The technical novelty is limited, as the dataset construction is entirely manual, and the resulting insight that public LLMs perform poorly in certain specialized domains is not new.

3. The authors are encouraged to discuss how the crafted dataset benefits the community. It is intuitive that LLMs struggle with unseen domains, so the benchmark results are unsurprising. Moreover, if the long-term goal is finetuning, a larger and more diverse dataset would be necessary.

**Questions:**

I have included my questions in the weakness section.

---

> ### Author Response · Authors · 2025-12-03
> **Response**
>
> We thank the reviewer for the detailed feedback and for recognizing Pluto as a benchmark that “fills a missing piece” in evaluating Verilog beyond correctness and toward efficiency. We address the concerns as follows:
>
> 1. **Value to the ICLR community.** We appreciate the reviewer’s perspective and clarify that, although Pluto is instantiated in hardware, the contributions of this work target core LLM capabilities that extend far beyond Verilog and are directly relevant to the ICLR community. The benchmark is not about demonstrating human-crafted HDL, but about exposing fundamental limitations in how LLMs reason about optimization, efficiency, and tradeoffs. Pluto is the first benchmark that requires LLMs to reason over competing objectives and produce code that lies on or near a Pareto front. This setting generalizes to any domain where solutions must balance tradeoffs. Additionally, our generalized eff@k formulation introduces a multidimensional efficiency metric; prior code-generation benchmarks evaluate only along a single dimension (i.e. runtime in software code) and cannot express or measure efficiency across multiple optimization targets.
>
> 2. **Limited Technical Novelty.** While the dataset itself is manually constructed, consistent with prior hardware-design benchmark, our core contributions extend beyond dataset curation and optimization. Pluto introduces new evaluation capabilities that did not exist in any prior work and that fundamentally change how LLM-generated hardware code can be assessed:
>    - **Per-metric Pareto-optimal reference designs.** Our benchmark includes per-metric Pareto-optimal reference designs, revealing a 10–22% efficiency gap in canonical RTLLM and VerilogEval solutions, and showing that prior baselines were suboptimal and not suitable for benchmarking LLMs on hardware code efficiency.
>
>    - **Clock-cycle–agnostic testbenches.** Our clock-cycle agnostic testbenches support variable-latency and optimization-dependent execution, which is an ability missing from all prior hardware benchmarks (VerilogEval and RTLLM). This technical contribution is essential for evaluating realistic hardware optimizations.
>
>    - **First benchmark for multi-objective hardware optimization.** Pluto is the first benchmark to evaluate multi-objective hardware optimization, enabling simultaneous assessment across area, delay, and power. In contrast, existing code-efficiency benchmarks for software (Enamel and Mercury) evaluate only along a single dimension: runtime.
>
>    Overall, Pluto introduces new tasks, new evaluation signals, and new insights that could not be obtained from existing datasets.
>
> 3. **How Pluto benefits the community?** The goal for Pluto is to provide a benchmark set to evaluate LLMs on optimizing different objectives in hardware design. Pluto is the first and only code optimization benchmark that contains three different code writing styles where each variant is optimized for a specific metric. Together, these three variants form a pareto-front which also extends pluto to evaluate LLMs in optimizing multiple competing objectives together (as discussed in Section 5 of the paper). The multi-objective optimization task and per-metric optimization tasks haven’t been explored in software coding environments as software coding benchmarks (Enamel and Mercury) only explored code efficiency in terms of one metric which is run-time. While a larger dataset would be appropriate for finetuning, Pluto’s value lies primarily in evaluation, it delivers high-quality, expert-curated ground truth for capabilities that are currently not available in other hardware coding benchmarks.

---

### Official Review · Reviewer_r52R · 2025-11-01

**Soundness:** 3
**Presentation:** 3
**Contribution:** 2
**Rating:** 4
**Confidence:** 4

**Summary:**

The paper is proposing a new dataset benchmark for testing Verilog code generation functionality. At the same time, this newly proposed benchmark identifies an area that previous datasets didn’t cover: PPA. The paper proposes a new dataset where each question has three golden standards optimized for power efficiency, delay, and area.

**Strengths:**

Built on previous datasets, the new benchmark this paper proposes includes PPA information for Skywater130HD and TSMC 65nm.
Both Cadence Genus and Yosys Wolf show similar optimization strength over the data, proving the results are valid.
Per-metric ground truth for PPA is provided in the benchmark.

**Weaknesses:**

1. While Figure 5(a) presents results analyzing the extent to which LLMs can achieve PPA optimization, the region corresponding to Low Functionality–High Efficiency lacks data points. For example, if a prompt requests generation of a 1024-bit adder but the LLM produces a 64-bit adder instead, the result would exhibit low functionality but high efficiency. A similar case arises if an ALU omits certain operations. Such designs can still be synthesizable and compatible with the testbench, provided appropriate pin mappings are added.

2. Although the paper proposes golden standards for each PPA metric based on the prior dataset’s questions, it overlooks an essential aspect. Defining golden standards is valuable, but achieving a balance among PPA metrics is even more critical. Consider a scenario where this benchmark is used for testing, and an LLM produces a module optimization that improves latency but deviates from the golden standard due to superior area performance—should this outcome still be considered a pass? The paper does not specify any quantitative method for handling such cases of “partial” or “balanced” optimization.

3. Furthermore, with 114 total tasks divided evenly into 38 per metric, the dataset remains relatively small. Its usefulness could be significantly enhanced by including hierarchical designs that integrate multiple modules.

4. Additionally, the absence of FPGA or ASIC implementation results is concerning; relying solely on simulation-based PPA evaluation may undermine result robustness.

5. Finally, the Pass@K explanation in the appendix is somewhat misleading. Instead of formally describing the combinatorial basis (n choose k), it presents three illustrative images for Pass@1, Pass@5, and Pass@10, which incorrectly suggest an iterative evaluation process.

**Questions:**

1. Why was low functionality and high performance never triggered?
2. Are there any designs included that are slightly more complex?
3. What score should we give an LLM when it makes an optimization that is balanced across multiple
metrics?
4. Other see weakness

---

> ### Author Response · Authors · 2025-12-03
> **Response Addressing Weaknesses (Part 1)**
>
> We thank the reviewer for their detailed feedback. We address each of the concerns as follows:
>
> 1. **Why is the low pass@k and high eff@k quadrant empty ?** This region corresponds to problems that fail functionality most of the time (low pass@k) but somehow produces highly efficient code. In practice, this scenario is very rare to happen because computing eff@k is dependent on the pass@k, the solution must pass functionality first to be evaluated on efficiency. Scenarios such as generating 64-bit adder instead of the requested 1024-bit version, or omitting operations from ALU implementation will potentially yield optimized solutions, however, such implementations should fail the functionality check. The adder testbench should include operand values exceeding 64-bits, and the ALU testbench should exercise all expected operations to ensure high coverage. Even if such cases were allowed into the evaluation due to partial crediting of functional correctness, they should not be considered “high-efficiency successes,” because their efficiency comes from incorrect or incomplete functionality, not meaningful optimization.
>
> 2. **Balanced PPA optimization.** Thank you for raising this important point. We agree that in practical design workflows, improvements in one metric may justify regressions in another. To address this, we have added Section 5 of the paper to introduce multi-objective evaluation,  where models are assessed relative to the appropriate Pareto front constructed from Pluto’s optimized reference implementations. Specifically, the benchmark evaluates solutions against either a two-objective (e.g area and delay or area and power or delay and power) or three-objective Pareto front (area, delay, and power), depending on the optimization target specified in the prompt.
>
> 3. **Dataset size clarification**. Pluto includes 114 different problems, and each problem includes three independently optimized canonical solutions: one optimized for area, one for delay, and one for power. This results in **342 Verilog implementations in total (3 × 114), in addition to one unoptimized baseline per problem**. We report 114 as the total number of tasks since each one of these three implementations share the same testbench and the same natural language specification and unoptimized baseline reference; they represent different optimization targets for the same task.
>
> 4. **FPGA and ASIC Implementation**. We agree that evaluating the PPA on real FPGA and ASIC implementations will provide an additional evaluation depth. In this work, we rely on post-synthesis PPA evaluation because it is widely used in evaluating tradeoffs between different RTL writing styles and microarchitecture choices. Once RTL implementation and architecture is fixed, backend implementation begins. At this stage, PPA optimizations rely mainly on layout optimizations (e.g. buffer insertion, cell swap, re-routing) and very rarely require RTL changes. Therefore, post-synthesis PPA provides a meaningful metric for comparing the relative performance of LLM generated RTL code.
>
> 5. **Pass@k explanation.** We have revised Appendix 4 to include a formal combinatorial explanation of the pass@k metric and clarifying that the metric is computed from a single set of n generated samples rather than through iterative evaluation. We have also better connected the sampling diversity examples to the metric's interpretation.

---

> ### Author Response · Authors · 2025-12-03
> **Response Addressing Questions (Part 2)**
>
> We clarify the raised questions below:
>
> 1. **Why was low functionality and high performance never triggered?** Answered in Point 1 in the previous comment.
> 2. **Are there any designs included that are slightly more complex?**
>    Yes, Pluto contains designs that span different complexity ranges, categorized into easy, medium, and hard.  These categories are determined based on optimization difficulty rather than raw circuit size. The table below summarizes the statistics for these three categories. We note that medium designs may be structurally larger, but hard designs are labeled “hard” because they are more difficult to optimize for PPA, not because they have more gates.
>
>    | **Difficulty** | **Avg # of IO Pins** | **Avg # of Synthesized Gates** | **Max # of Gates** |
>    |----------------|-----------------------|---------------------------------|----------------------|
>    | Easy           | 50                    | 181                             | 777                  |
>    | Medium         | 127                   | 629                             | 8815                 |
>    | Hard           | 79                    | 791                             | 3100                 |
>
> 3. **What score should we give an LLM when it makes an optimization that is balanced across multiple metrics?** Answered in Point 2 in the previous comment.

---

### Meta-Review · Area_Chair_KpFv · 2026-01-06

**Summary:**

This paper introduces Pluto, a benchmark and evaluation framework for assessing LLM-generated Verilog code not only on functional correctness, but crucially on hardware efficiency metrics: area, delay, and power (PPA).

**Reviewer Concerns:**

**Benchmark-Centric Contribution vs. ICLR Fit**

Multiple reviewers questioned whether Pluto provides sufficient methodological or algorithmic insight for the ICLR community.
Concerns that the paper primarily demonstrates that LLMs underperform in a specialized domain, which is not surprising.
Some reviewers suggested the work may be better suited for EDA venues (DAC/ICCAD).

**Reliance on Expert-Crafted Golden Designs**

Several reviewers questioned the scalability and necessity of manually crafted Pareto-optimal references.
Suggested alternatives such as relative scoring or predicted Pareto fronts to reduce expert effort.
Authors defended golden designs as essential anchors for true efficiency evaluation.
Dataset Size and Complexity

114 tasks seen as small, especially for future fine-tuning or broader generalization.
Requests for more complex or hierarchical designs.
Authors clarified complexity tiers and gate-count statistics, but concerns remained for some reviewers.

**Evaluation Design and Metrics**

- Empty “low functionality–high efficiency” region in results
- Handling of balanced or partial PPA improvements
- Definition and explanation of pass@k and eff@k

Some reviewers wanted deeper insight into why LLMs fail at optimization (e.g., structural patterns, FSM encoding, arithmetic fusion).
Authors expanded analysis and added variance/statistical reporting.

**Reviewer Scores:**

Reviewer r52R (Score: 4, borderline reject)
Focused on evaluation edge cases, dataset size, and ambiguity in balanced PPA scoring. Generally constructive; concerns were largely addressed in rebuttal.

Reviewer uhb7 (Score: 2, reject)
Strongest negative reviewer. Main argument: limited novelty and limited value to ICLR; dataset alone insufficient; insights unsurprising.

Reviewer MZkp (Score: 6, borderline accept)
Generally positive; asked for deeper failure analysis and statistical robustness. Rebuttal addressed most points.

Reviewer jfoZ (Score: 2, reject)
Skeptical of expert-crafted goldens, high confidence rejection.

In all, even with point raised from 4->5 and 6->7 the paper would still be on the rejection side.

---

### Decision · Program_Chairs · 2026-01-26

Reject